# Knizhnik-Zamolodchikov equations and integrable hyperbolic Landau-Zener models

Suvendu Barik[1*], Lieuwe Bakker[1†], Vladimir Gritsev[1,2‡] and Emil A. Yuzbashyan[3∘]

**1** Institute for Theoretical Physics, Universiteit van Amsterdam,
Science Park 904, 1098XH Amsterdam, The Netherlands
**2** Russian Quantum Center, Skolkovo, Moscow 143025, Russia
**3** Department of Physics and Astronomy, Center for Materials Theory,
Rutgers University, Piscataway, New Jersey 08854, USA

⋆ s.k.barik@uva.nl , † l.bakker@friam.nl , ‡ v.gritsev@uva.nl , ∘ eyuzbash@physics.rutgers.edu

## Abstract

We study the relationship between integrable Landau-Zener (LZ) models and Knizhnik-Zamolodchikov (KZ) equations. The latter are originally equations for the correlation functions of two-dimensional conformal field theories, but can also be interpreted as multi-time Schrödinger equations. The general LZ problem is to find probabilities of tunneling from eigenstates at $t = t_{\text{in}}$ to eigenstates at $t \to +\infty$ for an $N \times N$ time-dependent Hamiltonian $\hat{H}(t)$. A number of such problems are exactly solvable in the sense that their tunneling probabilities are elementary functions of Hamiltonian parameters. Recently, it has been proposed that exactly solvable LZ models of this type map to KZ equations. Here we use this connection to identify and solve a class of integrable LZ models with hyperbolic time dependence, $\hat{H}(t) = \hat{A} + \hat{B}/t$, for $N = 2, 3$, and $4$, where $\hat{A}$ and $\hat{B}$ are time-independent matrices.

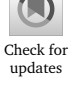

# 1 Introduction

Exploring the dynamics of avoided crossings of energy levels is a well known venture in physics. Famously, they were studied in the context of one-dimensional, slow atomic collisions by Landau [1] and intra-molecular level transitions by Zener [2] in 1932. In the same year, Majorana [3] extensively studied a spin-1/2 system in a varying magnetic field, while Stückelberg [4] utilized JWKB theory to solve the corresponding differential equations. All of them essentially set out to determine the probability of an initial state transitioning to another eigenstate of the Hamiltonian as a function of time. This probability is referred to as the 'transition probability' and was initially studied in the non-adiabatic evolution of two-level systems. The results obtained have proven to be of integral importance to many advancements. For instance, Majorana's work explained the 'holes' in the magneto-optical traps used in the first realisation of a BEC[1] [5].

The Landau-Zener-Stückelberg-Majorana problem (LZSM or LZ in short)[2] [6] is well known in the studies of ions and molecules placed within a time-varying field. To calculate transition probabilities in multi-state systems (i.e., systems with Hilbert space dimension $N > 2$), it is often sufficient to consider only the avoided crossings between pairs of instan-

---

[1]The question on how to avoid these 'Majorana holes' was resolved by Ketterle by simply 'plugging' the hole using a focused laser.

[2]In this work we will refer to this problem as the 'LZ problem' for brevity.

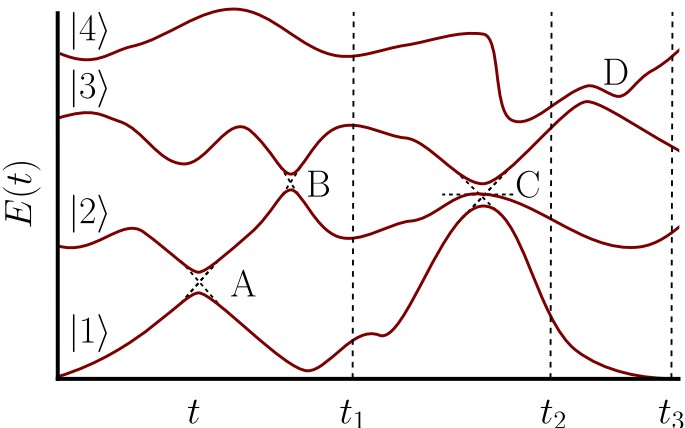

Figure 1: Schematic depiction of adiabatic (instantaneous) energy levels labelled by $|\alpha\rangle$ for $\alpha = \{1\ldots,4\}$. As an example, consider the transition probability $P_{|1\rangle \to |3\rangle}$ starting from $t = 0$. At $t_1$ it can be approximated by two standard $2 \times 2$ LZ problems defined at anticrossings A and B. The same probability at time $t_2$ requires an understanding of a higher order ($N = 3$) LZ problem at point C. Finding the probability at time $t_3$ requires the solution to a non-standard (i.e. nonlinear) LZ problem as depicted at point D.

taneous energy levels. This approach, known as the independent crossing approximation, estimates the total transition probability by treating each crossing as independent and approximating the overall evolution as a sequence of two-level transitions. Each individual transition is described by the linearised $2 \times 2$ LZ model, as originally solved by Landau, Zener, Stückelberg, and Majorana. A schematic description of the setting is given in Fig 1.

Not surprisingly this simplification down to two-level systems does not resolve the general setting of the problem. For example, when linear time dependence of the model is not sufficient (i.e. the level crossings cannot be linearised) or when the anti-crossing involves more than two energy levels, the transition probabilities cannot be readily calculated using the solutions found in 1932. Hence, other varieties of the LZ problem require consideration. In the 90 years after the original papers of 1932, an assortment of these problems have been addressed.

Known exactly solvable LZ models with linear time dependence include Demkov-Osherov model [7], where only a single diagonal matrix element is time-dependent in the diabatic basis, and the bow-tie and generalised bow-tie models [8–10], where all or most diabatic energy levels cross at a single point. A many-body example is the inhomogeneous Dicke model [11–14], which describes a collection of two-level systems—each with its own level splitting—interacting with a single bosonic mode whose frequency is a linear function of time. These models are exactly solvable in the sense that transition probabilities from $t = -\infty$ to $t = +\infty$ are found explicitly in terms of elementary functions.

A natural question to consider is; what is special about these and other similar LZ models that makes them exactly solvable? This question was addressed in [15] with the conclusion that the *necessary* condition for LZ solvability is the *quantum integrability* of the model, in the sense that there exist nontrivial, mutually commuting partners,

$$[\hat{H}_i(t), \hat{H}_j(t)] = 0, \quad i, j = 1, \ldots, n, \tag{1}$$

where $H_1(t) \equiv H(t)$ is the model (LZ) Hamiltonian and $H_i(t)$ with $i > 1$ are its commuting partners. With appropriate restrictions on $H_i(t)$ to make the condition (1) nontrivial, this is a good analogue [16–18] of classical *Liouville integrability* [19]. In the above examples, the $H_i(t)$ are required to be linear in $t$. For a nonlinear LZ model this requirement has to be generalised.

This quantum version of Liouville integrability is clearly only necessary and not sufficient for a LZ problem to be exactly solvable. Indeed, most quantum integrable models (e.g., the 1D Hubbard or XXZ Hamiltonians) do not turn into exactly solvable LZ models when we make their parameters (e.g., Hubbard $U$ or the anisotropy in the XXZ Hamiltonian) depend on time in an arbitrary way. It has been conjectured in [13] that a *sufficient* condition is the existence of a consistent set of multi-time Schrödinger equations

$$i\nu \frac{\partial \Psi(\vec{z})}{\partial z_i} = \hat{H}_i \Psi(\vec{z}), \qquad i = 1, \ldots, n. \tag{2}$$

Here $\vec{z} = (z_1, \ldots z_n)$ and $\nu$ are the parameters of the LZ model, one of which is the rescaled time variable, $z_1 \equiv \nu t$. In other words, the first equation in (2) is the nonstationary Schrödinger equation of the original LZ problem, which we are seeking to solve. The remaining $z_i$, which play the role of additional 'times' in equation (2), are other parameters of the underlying LZ Hamiltonian. For example, in the inhomogeneous Dicke model $z_i$ are the level splittings of the two-level systems while the energy of the bosonic mode $\omega$ changes at the rate $\nu$, i.e., $\omega = -\nu t$.

The system of multi-time Schrödinger equations (2) is compatible if and only if the *Frobenius integrability* condition is satisfied [20],

$$\frac{\partial \hat{H}_i}{\partial z_j} - \frac{\partial \hat{H}_j}{\partial z_i} - i[\hat{H}_i, \hat{H}_j] = 0, \quad i, j = 1, \ldots, n. \tag{3}$$

Note that for real Hamiltonians ($\hat{H}_i^* = \hat{H}_i$) the real and imaginary parts of equation (3) separate into two conditions, one of which is equation (1) while the other reads

$$\frac{\partial \hat{H}_i}{\partial z_j} = \frac{\partial \hat{H}_j}{\partial z_i}. \tag{4}$$

Thus, in this context the Frobenius integrability is more restrictive than the Liouville one.

Essentially the only nontrivial example of a multi-time system (2) we are aware of, such that $\hat{H}_i$ admit a representation in terms of finite matrices. are the Knizhnik-Zamolodchikov (KZ) equations and their various generalisations. The original KZ equations [21] are differential equations for $n$-point correlation functions $\Psi(\vec{z})$ in Wess-Zumino-Witten models. In this case, $\hat{H}_i$ in equation (2) have the following form:

$$\hat{H}_i = \sum_{j \neq i, \alpha, \beta} \frac{\eta_{\alpha\beta} \hat{r}_i^{\alpha} \otimes \hat{r}_j^{\beta}}{z_j - z_i}, \tag{5}$$

where $\hat{r}_i^{\alpha}$ are the generators of a Lie algebra and $\eta$ is its Killing form. These $\hat{H}_i$ are known as rational Gaudin magnets [22]. In addition, there are integrable hyperbolic, trigonometric, and elliptic Gaudin magnets [22, 23], where the 'couplings' $\frac{1}{z_i - z_j}$ are replaced by hyperbolic, trigonometric and elliptic functions of $(z_i - z_j)$ that are also different for different values of $\alpha$ and $\beta$ (anisotropic). Various boundary terms (terms that single out $\hat{r}_i^{\alpha}$) can be added to $\hat{H}_i$ without spoiling the integrability [23–29]. All these generalised Gaudin magnets satisfy the Frobenius integrability condition (3) and, therefore, give rise to integrable generalisations of KZ equations and, according to the above conjecture, to integrable LZ models.

Unfortunately, the solution to the KZ equations is extremely complicated. It is written in terms of a multidimensional contour integral over the so-called Yang-Yang action, derived from the off-shell Bethe Ansatz equations [30–33]. For a summary of results on the KZ equations, we refer to the Askey-Bateman Project: Volume 2 and references therein [34] as well as Aomoto's book on the cohomology methods in resolving such integrals [35]. As a result, applying the

general solution of the KZ equations in practice remains a complicated endeavour. That is not to say that this solution is not already useful. For example, Zabalo et al. [36] used the general solution to derive the long time asymptotic wavefunction for the spin-1/2 BCS Hamiltonian with the pairing strength inversely proportional to time for any system size through a saddle point approximation.

The purpose of the present paper is to initiate a systematic construction and explicit solution of integrable LZ models of physical interest, by utilizing their link to the KZ equations. Specifically, we derive here several simplest nontrivial examples of integrable LZ models from the KZ equations, solve them directly, and extract their solution from the contour integral solution of the KZ equations.

We focus on the case when it is impossible to describe LZ tunnelling by linearising the anticrossings. For example, the electromagnetic potentials that describe collisions of atoms and ions are inversely proportional to the radius. In this scenario, assuming a constant velocity, the LZ Hamiltonian is of the form $\hat{H}(t) = \hat{A} + \hat{B}/t$, where $\hat{A}$ and $\hat{B}$ are time-independent Hermitian operators. These type of problems were originally dubbed 'Coulomb' LZ problems, referring to the Coulomb potential involved [37–41]. Alternatively, these Coulomb models can be described by Nikitin's [42] exponential models through a simple transformation described in Section 3. We note that we prefer calling such LZ problems Hyperbolic LZ (HLZ) problems, to accommodate more general physical setups. There are plethora of reasons to study hyperbolic LZ models. Table III in [42] gives some nice examples such as the ion-atom collisions stated above and transitions between vibrational modes [43]. These problems also manifest in Rydberg transitions and molecular collisions. See [44] and Refs. [25-26] in [45] for detailed examples.

This work is structured as follows. In section 2, we start with an introduction to the KZ equations for the BCS (a.k.a. Richardson or Richardson-Gaudin) model with the superconducting coupling inversely proportional to time as discussed in [14] and [36]. We derive the HLZ models from this BCS problem and obtain their solutions from the solution to the KZ equations by means of the contour integration. The second part of the work in section 3 returns to the matter of solving the HLZ problems in terms of transition probabilities. This is done explicitly for the models considered in the first part as well as for their generalisations. For some of the models solved, the transition probabilities were obtained in [45], without fully solving the model. Here, however, full analytical solutions of the non-stationary Schrödinger equation for these models are presented, and the resulting transition probabilities are obtained. These results are derived by analytically solving the differential equations. It is also shown that these results coincide with the results presented in the first part of this work, verifying the contour-integral approach for these HLZ models. Finally, in section 4 we identify a number of new integrable multi-level HLZ models through the KZ connection.

We do not seek to fully solve all of these new models, but do derive several explicit exact solutions of their non-stationary Schrödinger equations as an example. These solutions are obtained through the exact solution provided by the KZ equations, after evaluating the contour integral. We conclude by discussing our results and outlining outstanding problems and topics of further interest in the Conclusion & Discussion.

## 2 Generalised KZ equations and the BCS Hamiltonian

### 2.1 Preliminaries

The generalised Knizhnik-Zamolodchikov equations we will use in this paper are of the form
[14, 25, 28, 29]

$$
\begin{aligned}
i\nu\frac{\partial\,|\Psi\rangle}{\partial\,\varepsilon_j} &= \hat{H}_j\,|\Psi\rangle\,, \qquad j = 1,\ldots,N\,, \\
i\nu\frac{\partial\,|\Psi\rangle}{\partial\,\Omega} &= \hat{H}_\Omega\,|\Psi\rangle\,,
\end{aligned}
\tag{6}
$$

where

$$
\hat{H}_j = 2\Omega\hat{s}_j^z - \sum_{k\neq j}\frac{\hat{\mathbf{s}}_j\cdot\hat{\mathbf{s}}_k}{\varepsilon_j - \varepsilon_k}\,, \qquad \hat{H}_\Omega = 2\sum_{j=1}^N \varepsilon_j\hat{s}_j^z - \frac{1}{2\Omega}\sum_{j,k}s_j^+ s_k^-\,.
\tag{7}
$$

Let $\Omega \equiv \Omega(t)$ be a function of time. Then, the second equation in (6) becomes the non-stationary Schrödinger equation for the time-dependent BCS (a.k.a. Richardson) Hamiltonian

$$
\hat{H}_{\text{BCS}}(t) \equiv \hat{H}_\Omega(t) = 2\nu^{-1}\dot{\Omega}\sum_j \varepsilon_j\hat{s}_j^z - (2\nu\Omega)^{-1}\dot{\Omega}\sum_{j,k}\hat{s}_j^+\hat{s}_k^-\,.
\tag{8}
$$

In this work $\Omega(t) = \nu t$, which yields a BCS Hamiltonian with the coupling inversely proportional to time:

$$
\hat{H}_{\text{BCS}}(t) = 2\sum_j \varepsilon_j\hat{s}_j^z - \frac{1}{2\nu t}\sum_{j,k}\hat{s}_j^+\hat{s}_k^-\,.
\tag{9}
$$

Note that $\hat{s}_j$ are general spin operators of arbitrary magnitude $s$. The parameters $\varepsilon_j$ play the role of on-site Zeeman magnetic fields. In the original fermion language, $\varepsilon_j$ are the single-particle energy levels [46–48]. We also reiterate the general form of the hyperbolic Landau-Zener (HLZ) model:

$$
i\partial_t\,|\Psi(t)\rangle = \left(\hat{A} + \frac{1}{t}\hat{B}\right)|\Psi(t)\rangle\,,
\tag{10}
$$

where $\hat{A}$ and $\hat{B}$ are constant, Hermitian matrices written in the diabatic basis defined through diagonalizing $\hat{B}$ by an orthogonal transformation. The evolution begins at $t \to 0^+$ and proceeds towards the positive direction of infinity. We now demonstrate that the BCS Hamiltonian (9) comprises many such HLZ problems.

#### 2.1.1 Two-site BCS models

In the basis where the $z$-component of the total spin is diagonal, the Hamiltonian (9) is block diagonal, with each block corresponding to a particular value of $S^z \in \{-sN,\ldots,sN\}$, i.e., $H_{\text{BCS}} = \bigoplus_{S^z=-sN}^{sN} H_{\text{BCS}}^{(S^z)}$. We note that, starting from this section, we add spin labels to operators that explicitly indicate the spin representations we are using.

For spin-1/2, we have

$$
\hat{s}_{1/2}^z = \frac{1}{2}\begin{pmatrix} 1 & 0 \\ 0 & -1 \end{pmatrix}\,, \qquad \hat{s}_{1/2}^+ = \begin{pmatrix} 0 & 1 \\ 0 & 0 \end{pmatrix}\,, \qquad \hat{s}_{1/2}^- = \begin{pmatrix} 0 & 0 \\ 1 & 0 \end{pmatrix}\,.
\tag{11}
$$

We can write $|\Psi(t)\rangle$ for $N = 2$ as

$$
|\Psi(t)\rangle = \sum_{i,j\in\{\uparrow,\downarrow\}} \psi_{i,j}(t)\,|i\rangle \otimes |j\rangle\,,
\tag{12}
$$

where $|i\rangle$ is the eigenstate of $\hat{s}^z_{1/2}$ in (11). Rewriting this state as a column vector with elements $\psi_{i,j}(t)$ with ordering of the $(i,j)$ indices as $[(\downarrow,\downarrow), (\uparrow,\downarrow), (\downarrow,\uparrow), (\uparrow,\uparrow)]$, we have

$$\hat{H}_{\text{BCS},1/2} = \begin{pmatrix} H^{(-1)}_{1/2} & \cdot & \cdot \\ \cdot & H^{(0)}_{1/2} & \cdot \\ \cdot & \cdot & H^{(1)}_{1/2} \end{pmatrix}, \tag{13}$$

where

$$H^{(-1)}_{1/2}(\nu) = -(\varepsilon_1 + \varepsilon_2), \tag{14a}$$

$$H^{(1)}_{1/2}(\nu) = (\varepsilon_1 + \varepsilon_2) - \frac{1}{\nu t}, \tag{14b}$$

$$H^{(0)}_{1/2}(\nu) = (\varepsilon_1 - \varepsilon_2)\sigma^z - \frac{1}{2\nu t}(\mathbb{I} + \sigma^x), \tag{14c}$$

and $\sigma^i$ are the usual Pauli matrices. Note that the $S^z = 0$ sector defines a '$2 \times 2$ HLZ problem'. Next, we consider spin-1 representations of the operators in (9):

$$\hat{s}^z_1 = \begin{pmatrix} 1 & 0 & 0 \\ 0 & 0 & 0 \\ 0 & 0 & -1 \end{pmatrix}, \qquad \hat{s}^+_1 = \sqrt{2}\begin{pmatrix} 0 & 1 & 0 \\ 0 & 0 & 1 \\ 0 & 0 & 0 \end{pmatrix}, \qquad \hat{s}^-_1 = \sqrt{2}\begin{pmatrix} 0 & 0 & 0 \\ 1 & 0 & 0 \\ 0 & 1 & 0 \end{pmatrix}. \tag{15}$$

Choose the basis $|i\rangle$, $i = \{0, \pm 1\}$ in (12) with the ordering of $(i,j)$ such that the ordering of the eigenstates is $[(-1,-1), (0,-1), (-1,0), (1,-1), (0,0), (-1,1), (1,0), (0,1), (1,1)]$. The Hamiltonian becomes

$$\hat{H}_{\text{BCS},1} = \begin{pmatrix} H^{(-2)}_1 & \cdot & \cdot & \cdot & \cdot \\ \cdot & H^{(-1)}_1 & \cdot & \cdot & \cdot \\ \cdot & \cdot & H^{(0)}_1 & \cdot & \cdot \\ \cdot & \cdot & \cdot & H^{(1)}_1 & \cdot \\ \cdot & \cdot & \cdot & \cdot & H^{(2)}_1 \end{pmatrix}. \tag{16}$$

Here we identify

$$H^{(\pm 2)}_1(\nu) = 2H^{(\pm 1)}_{1/2}(\nu), \tag{17a}$$

$$H^{(-1)}_1(\nu) = H^{(0)}_{1/2}(\nu/2) - (\varepsilon_1 + \varepsilon_2)\mathbb{I}, \tag{17b}$$

$$H^{(1)}_1(\nu) = H^{(0)}_{1/2}(\nu/2) + \left(\varepsilon_1 + \varepsilon_2 - \frac{1}{\nu t}\right)\mathbb{I}, \tag{17c}$$

$$H^{(0)}_1(\nu) = 2(\varepsilon_1 - \varepsilon_2)\hat{s}^z_1 - \frac{1}{\nu t}\left(2\mathbb{I} + \sqrt{2}\hat{s}^x_1 - (\hat{s}^z_1)^2\right). \tag{17d}$$

We notice that the $S^z = \pm 1$ sector, up to a rescaling of $\nu$ and adding a multiple of the $2 \times 2$ identity matrix, is nothing but the $S^z = 0$ sector from the spin-1/2 model. Then the only novel part that appears in equation (17) is the $S^z = 0$ sector. This sector is identified as a '$3 \times 3$ HLZ model'. After rotating to the following basis:

$$|1,0\rangle_g \equiv |1\rangle = \frac{1}{\sqrt{3}}\left(|-1,1\rangle + |0,0\rangle + |1,-1\rangle\right),$$

$$|2\rangle = \frac{1}{\sqrt{2}}\left(|1,-1\rangle - |-1,1\rangle\right), \tag{18}$$

$$|3\rangle = \frac{1}{\sqrt{3}}\left(-|-1,1\rangle + |0,0\rangle - |1,-1\rangle\right),$$

and denoting $\Delta = \varepsilon_2 - \varepsilon_1$ with $\varepsilon_2 > \varepsilon_1$, it transforms to a tridiagonal matrix, which we will discuss in Sec. 3.

The prescription outlined above can be carried out for general spin-$s$. The $S^z \neq 0$ magnetization sectors can always be written as a problem of a lower-spin BCS model up to a rescaling of the diabatic energy levels, while the $S^z = 0$ problem introduces additional complexity. As an example of this, we also provide the description of the spin-3/2 case in Sec. 4.

Similarly, one can also consider systems with more than two sites. For instance, the model in (74) is the $S^z = -1/2$ problem of a three-site spin 1/2 BCS model. One can also find varieties of HLZ problems by tweaking spin representations and the number of sites. As an illustration, we consider two and three site BCS models with different spins on each site in Appendix C.

The connection between HLZ problems and the KZ equations suggests that solving one problem can help understand the other. To this end, we wish to make sense of the solution for the LZ problem by means of the contour integral solution of the KZ equations, which we will do in the following subsections. In particular, we evaluate the contour integral for the spin 1/2 problem, compare the results with the brute force solution of the non-stationary Schrödinger equation in Sec. 3.1, and show that the resulting wavefunctions are indeed the same. Interestingly, but not surprisingly, it turns out that the choice of the contour determines the initial condition of the BCS and therefore the HLZ problems.

### 2.1.2 Integral representation of the solution of the KZ equations

As mentioned before, the generalised KZ equations (6) have an exact solution via an off-shell Bethe Ansatz [14, 25, 28–30]. For $N$ spins of lengths $s_j$ and the z-projection of the total spin $S^z = M - \sum_j^N s_j$, the solution is given by:

$$|\Psi(\Omega, \varepsilon)\rangle = \oint_\gamma d\lambda \exp\left[-\frac{i\mathcal{S}(\lambda, \varepsilon)}{\nu}\right] |\Phi(\lambda, \varepsilon)\rangle, \qquad d\lambda = \prod_{\alpha=1}^M d\lambda_\alpha, \tag{19}$$

where $\varepsilon = (\varepsilon_1, \ldots, \varepsilon_N)$ with $\varepsilon_1 < \varepsilon_2 < \ldots < \varepsilon_N$, $\lambda = (\lambda_1, \ldots, \lambda_M)$,

$$\begin{aligned}
\mathcal{S}(\lambda, \varepsilon) = &-2\Omega \sum_j \varepsilon_j s_j + 2\Omega \sum_\alpha \lambda_\alpha - \frac{1}{2} \sum_j \sum_{j \neq k} s_j s_k \ln(\varepsilon_j - \varepsilon_k) \\
&+ \sum_j \sum_\alpha s_j \ln(\varepsilon_j - \lambda_\alpha) - \frac{1}{2} \sum_\alpha \sum_{\beta \neq \alpha} \ln(\lambda_\beta - \lambda_\alpha),
\end{aligned} \tag{20}$$

and

$$|\Phi(\lambda, \varepsilon)\rangle = \prod_{\alpha=1}^M \hat{L}^+(\lambda_\alpha) |0\rangle, \qquad \hat{L}^+ = \sum_{j=1}^N \frac{\hat{s}_j^+}{\lambda - \varepsilon_j}. \tag{21}$$

The minimal weight state $|0\rangle$ is the state where all spins point in the negative z-direction, $\hat{s}_j^z |0\rangle = -s_j |0\rangle$. The closed contour $\gamma$ is such that the integrand comes back to its initial value after $\lambda_\alpha$ has described it.

The $s = 1/2$, $S^z = 0$ block of the Hamiltonian[3] (14c) can be solved in the following way. First, the Yang-Yang action (20) takes the explicit form

$$\begin{aligned}
\mathcal{S}(\lambda, \varepsilon) = &-\nu t(\varepsilon_1 + \varepsilon_2) + 2\nu t \lambda - \frac{1}{4} \log(\varepsilon_2 - \varepsilon_1) \\
&+ \frac{1}{2} \log(\varepsilon_1 - \lambda) + \frac{1}{2} \log(\varepsilon_2 - \lambda),
\end{aligned} \tag{22}$$

---

[3]We note that the $1 \times 1$ block can also be solved using the Yang-Yang action, although the result is trivial. For completeness, we provide this calculation in Appendix A.

where we dropped an imaginary constant that arises when we combine the $\log(\varepsilon_2 - \varepsilon_1)$ and $\log(\varepsilon_1 - \varepsilon_2)$ terms keeping in mind that $\varepsilon_2 > \varepsilon_1$. This constant is absorbed into the overall normalization factor $C$ independent of $t$, $\varepsilon_1$ and $\varepsilon_2$. The state $|\Phi(\lambda, \varepsilon)\rangle$ is given by

$$|\Phi(\lambda, \varepsilon)\rangle = \hat{L}^+(\lambda)|\downarrow\downarrow\rangle = \frac{1}{\lambda - \varepsilon_1}|\uparrow\downarrow\rangle + \frac{1}{\lambda - \varepsilon_2}|\downarrow\uparrow\rangle \,. \tag{23}$$

The solution then becomes

$$\begin{aligned}
|\Psi(t, \varepsilon)\rangle = {} & C e^{-\frac{i}{\nu}\left[-\nu t(\varepsilon_1 + \varepsilon_2) - \frac{1}{4}\log(\varepsilon_2 - \varepsilon_1)\right]} \\
& \times \oint_\gamma d\lambda \, e^{-\frac{i}{\nu}\left[2\nu t\lambda + \frac{1}{2}\log(\varepsilon_1 - \lambda) + \frac{1}{2}\log(\varepsilon_2 - \lambda)\right]} \left(\frac{1}{\lambda - \varepsilon_1}|\uparrow\downarrow\rangle + \frac{1}{\lambda - \varepsilon_2}|\downarrow\uparrow\rangle\right).
\end{aligned} \tag{24}$$

This simplifies to

$$\begin{aligned}
|\Psi(t, \varepsilon)\rangle = {} & C e^{it(\varepsilon_1 + \varepsilon_2)}(\varepsilon_2 - \varepsilon_1)^{\frac{i}{4\nu}} \left[\oint_\gamma d\lambda \, e^{-2it\lambda}(\varepsilon_1 - \lambda)^{-\frac{i}{2\nu} - 1}(\varepsilon_2 - \lambda)^{-\frac{i}{2\nu}}|\uparrow\downarrow\rangle \right. \\
& \left. + \oint_\gamma d\lambda \, e^{-2it\lambda}(\varepsilon_1 - \lambda)^{-\frac{i}{2\nu}}(\varepsilon_2 - \lambda)^{-\frac{i}{2\nu} - 1}|\downarrow\uparrow\rangle\right].
\end{aligned} \tag{25}$$

We then introduce a new variable $\eta$ defined by

$$\lambda = \frac{\varepsilon_2 + \varepsilon_1}{2} - \eta \frac{\varepsilon_2 - \varepsilon_1}{2} \,. \tag{26}$$

The integral now becomes

$$|\Psi(t, \Delta)\rangle = C(\Delta)^{-\frac{3i}{4\nu}}\left[\oint_\gamma d\eta \, e^{i\eta\Delta t}\frac{(1 - \eta^2)^{-\frac{i}{2\nu}}}{1 - \eta}|\uparrow\downarrow\rangle - \oint_\gamma d\eta \, e^{i\eta\Delta t}\frac{(1 - \eta^2)^{-\frac{i}{2\nu}}}{1 + \eta}|\downarrow\uparrow\rangle\right], \tag{27}$$

where we used $\Delta \equiv (\varepsilon_2 - \varepsilon_1)$ and collected all constants depending on $\nu$ only in the normalization $C$. Note that there is now a minus sign between the two integrals. Rotating to the following basis:

$$\begin{aligned}
|1/2, 0\rangle_g \equiv |1\rangle &= \frac{1}{\sqrt{2}}(|\uparrow\downarrow\rangle + |\downarrow\uparrow\rangle), \\
|2\rangle &= \frac{1}{\sqrt{2}}(|\uparrow\downarrow\rangle - |\downarrow\uparrow\rangle),
\end{aligned} \tag{28}$$

we reduce the integral to the following form:

$$\begin{aligned}
|\Psi(t, \Delta)\rangle = {} & C(\Delta)^{-\frac{3i}{4\nu}}\sqrt{2}\left[\oint_\gamma d\eta \, e^{i\eta\Delta t}\eta(\eta^2 - 1)^{-\frac{i}{2\nu} - 1}|1\rangle \right. \\
& \left. + \oint_\gamma d\eta \, e^{i\eta\Delta t}(\eta^2 - 1)^{-\frac{i}{2\nu} - 1}|2\rangle\right].
\end{aligned} \tag{29}$$

Here, we extracted an overall constant $(-1)^{-\frac{i}{2\nu} - 1}\exp\left[(-\frac{i}{2\nu} - 1)2\pi i r\right]$ for $r \in \mathbb{N}$. This function is multivalued, but since it affects only the overall prefactor, we can safely absorb this term into the normalization constant. For the first integral we use

$$d\eta = \frac{d(\eta^2 - 1)^{-\frac{i}{2\nu}}}{\frac{-i\eta}{\nu}(\eta^2 - 1)^{-\frac{i}{2\nu} - 1}} \,. \tag{30}$$

We then integrate by parts and use the fact that the boundary term vanishes. This finally leaves us with

$$|\Psi(t,\Delta)\rangle = C\,(\Delta)^{-\frac{3i}{4\nu}}\sqrt{2}\left[\nu t\Delta\oint_{\gamma}d\eta\,e^{i\eta\Delta t}\left(\eta^2-1\right)^{-\frac{i}{2\nu}}|1\rangle + \oint_{\gamma}d\eta\,e^{i\eta\Delta t}\left(\eta^2-1\right)^{-\frac{i}{2\nu}-1}|2\rangle\right].\quad(31)$$

It turns out that the solution to this integral is given by integral representations of the Bessel function of the first kind as found by Hänkel [49].

As alluded to earlier, the specific choice for the contour determines the initial condition of the system. The contours in question are shown in Figure 2. First, we consider the contour $\gamma_1$ in the left of Figure 2, which, as we will show in Sec. 3.1.1, corresponds to the solution where we start in the ground state. The solution to the integral (31) is then provided using the following result:

$$2\pi i J_\kappa(\tau) = \frac{1}{\sqrt{\pi}}e^{i3\kappa\pi}\left(\frac{\tau}{2}\right)^{-\kappa}\Gamma\left(\kappa+\frac{1}{2}\right)\oint_{\gamma_1}d\eta(\eta^2-1)^{-\kappa-\frac{1}{2}}e^{i\eta\tau}.\quad(32)$$

We identify $\tau = t\Delta$, and use $\Gamma(z+1) = z\Gamma(z)$, to write our final answer to the integral (31):

$$|\psi^1_{1/2,0}(\tau)\rangle = C\,(\Delta)^{-\frac{3i}{4\nu}}i2^{\frac{i}{2\nu}+2}\frac{\pi^{3/2}\nu\tau^{\frac{i}{2\nu}+\frac{1}{2}}}{e^{-\frac{3}{2\nu}}\Gamma\left(\frac{i}{2\nu}\right)}\left[-iJ_{\frac{i}{2\nu}-\frac{1}{2}}(\tau)|1\rangle + J_{\frac{i}{2\nu}+\frac{1}{2}}(\tau)|2\rangle\right].\quad(33)$$

The superscript 1 on $\psi$ indicates that the initial condition at $t=0$ is the ground state. We apply the superscript $n$ to denote the wavefunction evolving from the $(n-1)^{\text{th}}$ excited state. The subscripts $1/2$ and $0$ on $\psi$ describe the spin magnitude $s$ and the spin sector $S^z$ in the corresponding BCS problem. For a different initial condition (i.e. starting in the excited state $|2\rangle$ at $t=0^+$), one can use the contour $\gamma_2$ as given in the right in Figure 2. The integral in this case is solved by using

$$2\pi i J_\kappa(\tau) = \frac{1}{\sqrt{\pi}}\left(\frac{\tau}{2}\right)^{\kappa}\Gamma\left(\frac{1}{2}-\kappa\right)\oint_{\gamma_2}d\eta(\eta^2-1)^{\kappa-\frac{1}{2}}e^{i\eta\tau}.\quad(34)$$

We find

$$|\psi^2_{1/2,0}(\tau)\rangle = C(\Delta)^{-\frac{3i}{4\nu}}\frac{2^{-\frac{i}{2\nu}+2}\pi^{3/2}(-i)\nu\tau^{\frac{i}{2\nu}+\frac{1}{2}}}{\Gamma\left(\frac{i}{2\nu}\right)}\left[iJ_{-\frac{i}{2\nu}+\frac{1}{2}}(\tau)|1\rangle + J_{-\frac{i}{2\nu}-\frac{1}{2}}(\tau)|2\rangle\right].\quad(35)$$

## 2.2 Higher level models and the choice of contour

This subsection shows how more complex HLZ models can be derived and solved using the KZ equations and their contour integral solution.

### 2.2.1 Identifying contours

We start by justifying the two choices of the contour we made above for the simplest $2\times 2$ case. In Sec. 3.1.1, we show that these two contours correspond to the initial state being the ground state and excited state at $t\to 0^+$.

The choices of the contour in Fig. 2 can be understood by inspecting the stationary points of the Yang-Yang action $S(\lambda,\varepsilon)$. The stationary point equations, $\partial S/\partial\lambda_i = 0$, and the values of $\lambda_i$ that solve these equations are known as the Richardson-Gaudin equations and Richardson parameters [14,30] or, more generally, as Bethe equations and Bethe roots, respectively. These values of $\lambda_i$ determine the stationary states of the BCS Hamiltonian.

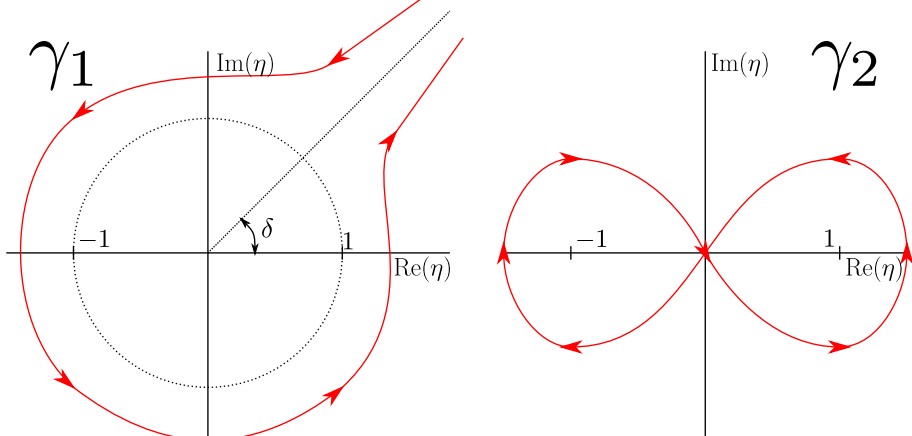

Figure 2: Two choices of the contour to solve equation (31). The red line and arrows draw the path and direction of the contours. The left contour, $\gamma_1$ corresponding to (32), encloses the unit circle. The contour is chosen such that $\delta \leq \arg(\eta) \leq 2\pi + \delta$, $-\delta \leq \tau \leq \pi - \delta$ and the values of $\eta$ range from $\delta$ to $\delta + \pi$. For $\gamma_2$ corresponding to (34), we only have the requirement that $\kappa + 1/2 \notin \mathbb{N}$.

The equation for the stationary points of the Yang-Yang action (22) reads

$$4\nu t = \frac{1}{\varepsilon_1 - \lambda} + \frac{1}{\varepsilon_2 - \lambda}. \tag{36}$$

The solutions of this equation in the limits $\nu t \to 0^+$ and $\nu t \to \infty$ are given by:

$$\nu t \to 0^+ \ : \ \begin{cases} \lambda \to -\infty, \\ \lambda \to \frac{\varepsilon_1 + \varepsilon_2}{2}, \end{cases} \qquad \nu t \to \infty \ : \ \begin{cases} \lambda \to \varepsilon_1, \\ \lambda \to \varepsilon_2. \end{cases} \tag{37}$$

The upper row of solutions corresponds to the lowest value for the Bethe root in (36) and the lower two solutions to the highest value for the Bethe root. From the exact solution (20), it can be seen that the evolution is dominated by the stationary points in the $\nu \to 0$ limit, and is adiabatic. Therefore, in order to start in the ground state with the lowest value for the Bethe root at $t = 0^+$, we presumably need a contour that can be pushed onto the point $\lambda = -\infty$, but not the point $\lambda = \frac{\varepsilon_1 + \varepsilon_2}{2}$ and vice versa for when the initial condition is the excited state. This argument can be made directly for the integrals solved in this section, however it is not immediately clear how to generalise it for more complicated integral solutions to the LZ problems.

### 2.2.2 Evolution from the ground state of a three-site model

There is a class of HLZ models which involves only a single contour in their integral solutions. They are identified from any $N$-site spin-$s$ BCS Hamiltonian by considering its $S^z = -sN + 1$ block. This scenario involves $N$ singular points on a complex plane. Here we focus on a three-site $s = 1/2$, $S^z = -1/2$ problem. We have

$$H_{1/2}^{(-3 \times 1/2 + 1)} = \sum_{i=1}^{3} \varepsilon_i \mathbb{I}_{3 \times 3} - 2 \begin{pmatrix} \varepsilon_1 & 0 & 0 \\ 0 & \varepsilon_2 & 0 \\ 0 & 0 & \varepsilon_3 \end{pmatrix} - \frac{1}{2\nu t} \begin{pmatrix} 1 & 1 & 1 \\ 1 & 1 & 1 \\ 1 & 1 & 1 \end{pmatrix}. \tag{38}$$

The Yang-Yang action for this problem is given by

$$S(\lambda, \varepsilon) = -\nu t(\varepsilon_1 + \varepsilon_2 + \varepsilon_3) + 2\nu t \lambda - \frac{1}{8} \sum_{i \neq j}^{3} \log(\varepsilon_i - \varepsilon_j) + \frac{1}{2} \sum_{i=1}^{3} \log(\varepsilon_i - \lambda). \tag{39}$$

The stationary points of the above action are determined by the following equation:

$$4vt = \sum_{i=1}^{3} \frac{1}{\varepsilon_i - \lambda}.\tag{40}$$

As in the previous subsection, we argue that for the lowest value of the Bethe root, the contour should begin and close at $\lambda \to -\infty$. It then represents the ground state of the model. This contour looks exactly as the $\gamma_1$ contour in Fig. 2, with all singular points inside it. Setting $\varepsilon_1 < \varepsilon_2 < \varepsilon_3$ in the Yang-Yang action and absorbing all constant terms into an overall normalization $C$, we obtain the integral representation of the solution $|\Psi(t,\varepsilon)\rangle$ as

$$
\begin{aligned}
|\Psi(t,\varepsilon)\rangle = C\Bigg[ &\oint_\gamma d\lambda\, e^{-2it\lambda} (\varepsilon_1-\lambda)^{-\frac{i}{2\nu}-1}(\varepsilon_2-\lambda)^{-\frac{i}{2\nu}}(\varepsilon_3-\lambda)^{-\frac{i}{2\nu}}|\uparrow\downarrow\downarrow\rangle \\
&+ \oint_\gamma d\lambda\, e^{-2it\lambda}(\varepsilon_1-\lambda)^{-\frac{i}{2\nu}}(\varepsilon_2-\lambda)^{-\frac{i}{2\nu}-1}(\varepsilon_3-\lambda)^{-\frac{i}{2\nu}}|\downarrow\uparrow\downarrow\rangle \\
&+ \oint_\gamma d\lambda\, e^{-2it\lambda}(\varepsilon_1-\lambda)^{-\frac{i}{2\nu}}(\varepsilon_2-\lambda)^{-\frac{i}{2\nu}}(\varepsilon_3-\lambda)^{-\frac{i}{2\nu}-1}|\downarrow\downarrow\uparrow\rangle\Bigg].
\end{aligned}\tag{41}
$$

As before, introducing a new variable $\eta$

$$\lambda = \varepsilon_2 - \eta(\varepsilon_3 - \varepsilon_1),\tag{42}$$

rotating to the basis

$$
\begin{aligned}
|1\rangle &= \frac{1}{\sqrt{3}}\left(|\uparrow\downarrow\downarrow\rangle + |\downarrow\uparrow\downarrow\rangle + |\downarrow\downarrow\uparrow\rangle\right), \\
|2\rangle &= \frac{1}{\sqrt{2}}\left(|\uparrow\downarrow\downarrow\rangle - |\downarrow\downarrow\uparrow\rangle\right), \\
|3\rangle &= \frac{1}{\sqrt{6}}\left(|\uparrow\downarrow\downarrow\rangle - 2|\downarrow\uparrow\downarrow\rangle + |\downarrow\downarrow\uparrow\rangle\right),
\end{aligned}\tag{43}
$$

and using $\Delta_{ij} \equiv \varepsilon_i - \varepsilon_j$, we reduce the integral to

$$|\Psi(t,\varepsilon)\rangle = C\Delta_{31}^{-\frac{3i}{2\nu}}e^{-2it\varepsilon_2}\oint_{\gamma_1}d\eta\, e^{2i\Delta_{31}t\eta}(\eta-r_1)^{-\frac{i}{2\nu}-1}(\eta+r_2)^{-\frac{i}{2\nu}-1}\eta^{-\frac{i}{2\nu}-1}\tag{44}$$

$$\times\left[-\frac{1}{\sqrt{3}}\left(3\eta^2 + 2(r_2-r_1)\eta - r_1r_2\right)|1\rangle + \frac{1}{\sqrt{2}}\eta|2\rangle - \frac{1}{\sqrt{6}}((r_2-r_1)\eta - 2r_1r_2)|3\rangle\right].$$

Here $r_1 = \Delta_{21}/\Delta_{31}$ and $r_2 = \Delta_{32}/\Delta_{31} = 1 - r_1$. This integral ultimately requires us to solve

$$I^\alpha = \oint_{\gamma_1} d\eta\, e^{2i\Delta_{31}t\eta}(\eta-r_1)^{-\frac{i}{2\nu}-1}(\eta+r_2)^{-\frac{i}{2\nu}-1}\eta^{-\frac{i}{2\nu}-1+\alpha}.\tag{45}$$

We tackle the above integral by using the Hänkel representation of the Gamma function. Relegating the details of the evaluation to Appendix A.2, we provide the closed expression for the integral (45):

$$I^\alpha = \frac{2\pi i\,\Gamma(3\omega-\alpha)^{-1}}{(2\Delta_{31}t)^{\alpha+1-3\omega}}F_{1:0;0}^{0:1;1}\left[\begin{array}{c}-\!\!-\!\!- : \omega;\omega \\ 3\omega-\alpha :-;-\end{array}\middle|\begin{array}{c}2ir_1\Delta_{31}t \\ -2ir_2\Delta_{31}t\end{array}\right],\tag{46}$$

where $F_{1:0;0}^{0:1;1}$ is a Kampé De Fériet function [50], which is a two-variable generalisation of the hypergeometric function, and $\omega = 1 + i/(2\nu)$. For the case where $\varepsilon_2 = \frac{1}{2}(\varepsilon_1 + \varepsilon_3)$, the above expression simplifies to a $_1F_2$ hypergeometric function,

$$I_{r_1 = r_2}^\alpha = \frac{2\pi i \, \Gamma(3\omega - \alpha)^{-1}}{(4i\Delta_{21}t)^{\alpha + 1 - 3\omega}} {}_1F_2\left[ {}_{\frac{3\omega}{2} - \frac{\alpha}{2}} \; {}_{\frac{3\omega}{2} - \frac{\alpha}{2} + \frac{1}{2}}^{\frac{\omega}{2}}; -(\Delta_{21}t)^2 \right]. \tag{47}$$

Finally, we substitute the above closed expressions into (44) to obtain the solution of the non-stationary Schrödinger equation for the Hamiltonian (38) for the evolution starting from the ground state. For conciseness, we do not provide the full expression here. A similar exercise is also done for a four-site $s = 1/2$, $S^z = -1$ problem which leads to a three-variable generalisation of the hypergeometric function [51]. Details are provided in Appendix A.3.

### 2.2.3 Multiple contours

We also point out an interesting example of a complicated integral structure that is found in the two-site spin-1 BCS Hamiltonian. Specifically, the Yang-Yang action corresponding to the $H_1^0$ sector in (17d) reads

$$\begin{aligned}
\mathcal{S}_{H_1^0}(\lambda, \varepsilon) = {} & -2\nu t(\varepsilon_1 + \varepsilon_2) + 2\nu t(\lambda_1 + \lambda_2) - \log(\varepsilon_2 - \varepsilon_1) + \log(\varepsilon_1 - \lambda_1) \\
& + \log(\varepsilon_2 - \lambda_1) + \log(\varepsilon_1 - \lambda_2) + \log(\varepsilon_2 - \lambda_2) - \log(\lambda_2 - \lambda_1).
\end{aligned} \tag{48}$$

Here, there are two integration variables, $\lambda_1$ and $\lambda_2$. This means that the solution as given by the Yang-Yang action (20) is a double contour integral. Unfortunately, we have not been able to perform this integral explicitly. However, from the corresponding HLZ problem and its solution (see Sec. 3.1) we know that the following must hold:

$$U \oint_\gamma d\lambda_1 d\lambda_2 \exp\left[ \frac{-i\mathcal{S}_{H_1^0}(\lambda, \varepsilon)}{\nu} \right] |\Phi(\lambda, \varepsilon)\rangle \propto |\psi_{1,0}^k(\tau)\rangle, \tag{49}$$

where $\varepsilon = (\varepsilon_1, \varepsilon_2)$, $\lambda = (\lambda_1, \lambda_2)$, $|\psi_{1,0}^k(\tau)\rangle$ are given in (68) and (B.1), and $U$ is the unitary transformation to the basis (18). The contours $\gamma$ in (49) determine which state $|\psi_{1,0}^k(\tau)\rangle$ is computed. We speculate that the choices for $\gamma$ are double contours comprised of the ones shown in Figure 2.

For example, we expect the initial condition where the HLZ problem (17d) starts in the ground state $|1\rangle$ in (18) to be associated with the contour $\gamma_1$ in Figure 2 with a second contour of the same shape enveloping the first. Starting in the first excited state then corresponds to $\gamma_1$ enveloping $\gamma_2$, and starting in the state $|3\rangle$ corresponds to a double $\gamma_2$ contour. We find that the number of possible combinations of the two contours given in Figure 2 equals the number of initial conditions for any spin-$s$ two-site BCS-derived HLZ problem. This no longer holds for BCS models with more than two sites due to additional branch points that appear in the integral representation of the solution to the corresponding KZ equations. The appropriate contours for these HLZ models are a topic of further investigation.

## 3 Hyperbolic Landau-Zener models

For the second part of the paper we demonstrate that the contour integral solutions of the KZ equations directly correspond to explicit solutions to differential equations for the HLZ models. The solution to the HLZ models presented here provide an important step towards understanding the aforementioned choice of contours and, by extension, a larger class of time-dependent quantum systems.

First, we note that under the substitution $t = e^w$, the differential equation (10) transforms into the form:

$$i\partial_w |\Psi(w)\rangle = \left[\hat{B} + e^w \hat{A}\right] |\Psi(w)\rangle \,, \tag{50}$$

where $w \to -\infty$ is equivalent to $t \to 0^+$ and $w \to \infty$ to $t \to \infty$. This shows that one can transform any exponential LZ problem to an HLZ model through a simple substitution [39].

After rewriting Eq. (10) in the diabatic basis, one of the diagonal terms can be eliminated by factoring out a global phase with trivial time dependence from the wavefunction. The resulting lowest non-trivial representation of the problem, assuming real-symmetric matrices, is then given by:

$$i\begin{pmatrix} \dot{\psi}_1(t) \\ \dot{\psi}_2(t) \end{pmatrix} = \begin{pmatrix} \frac{p}{t} + a_1 & a_2 \\ a_2 & 0 \end{pmatrix} \begin{pmatrix} \psi_1(t) \\ \psi_2(t) \end{pmatrix}. \tag{51}$$

This problem is the general real-symmetric $2 \times 2$ HLZ problem. After a time-independent rotation that transforms $x \to z$ and $z \to -x$, Eq. (14c) is seen to be a particular case of this problem. While the HLZ problem (51) has been extensively studied in the literature since the 1970s [37–41, 45], we provide a general solution in this work.

The most general $3 \times 3$ HLZ problem is, as far as we know, not solvable in terms of known special functions. The same applies to the general $3 \times 3$ LZ problem linear in $t$. However, as alluded to before, there turns out to be a particular version of the $3 \times 3$ HLZ problem, derived from the time-dependent BCS Hamiltonian (9), that is solvable:

$$i\begin{pmatrix} \dot{\psi}_1(t) \\ \dot{\psi}_2(t) \\ \dot{\psi}_3(t) \end{pmatrix} = \begin{pmatrix} \frac{p}{t} & a_1 & 0 \\ a_1 & \frac{q}{t} & a_2 \\ 0 & a_2 & 0 \end{pmatrix} \begin{pmatrix} \psi_1(t) \\ \psi_2(t) \\ \psi_3(t) \end{pmatrix}. \tag{52}$$

While this model has been partially addressed previously [45], the general solution of (52) provided in this work is new. We note that (51) and (52) are solvable for general choices of parameters $\{p, q, a_1, a_2\}$.

The goal of the LZ problem is to calculate the aptly named transition probability matrix. For an $N \times N$ problem, this matrix is written as

$$P_{N \times N} = \begin{pmatrix} p_{1 \to 1} & p_{1 \to 2} & \cdots & p_{1 \to N} \\ p_{2 \to 1} & p_{2 \to 2} & \cdots & p_{2 \to N} \\ \vdots & \vdots & \ddots & \vdots \\ p_{N \to 1} & p_{N \to 2} & \cdots & p_{N \to N} \end{pmatrix}. \tag{53}$$

Here, 1 refers to the ground state, 2 to the first excited state and so forth, up to $N$ being the highest excited state. $p_{m \to n}$ is the probability for the system starting in the $(m)^{\text{th}}$ state at the initial time to end up in the $(n)^{\text{th}}$ state at the final time. As mentioned earlier, for the HLZ problems presented here, the initial and final times are $t = 0^+$ and $t \to \infty$, respectively.

## 3.1 Solutions to differential equations

We now outline how the solutions to Eqs. (51) and (52) are obtained. Our main strategy is to reduce the system of coupled linear differential equations to a single higher-order ordinary differential equation. Once this equation is solved, the remaining components of the solution can be determined straightforwardly from it.

### 3.1.1 The general 2 × 2 HLZ problem

Eliminating $\psi_1(t)$ from the first equation in (51), we obtain,

$$t\ddot{\psi}_2(t) + i\left(p + a_1 t\right)\dot{\psi}_2(t) + a_2^2 t\psi_2(t) = 0, \tag{54}$$

$$\psi_1(t) = \frac{i}{a_2}\dot{\psi}_2(t). \tag{55}$$

The main differential equation of the 2×2 problem is Eq. (54) for $\psi_2(t)$, whose solutions then determine the remaining component $\psi_1(t)$. Substituting $\psi_2(t) = \exp\left(-it\left(a_1 + \mu\right)/2\right)g(t)$ and using $x = i\mu t$, with $\mu = \sqrt{a_1^2 + 4a_2^2}$, we rewrite (54) as

$$x\ddot{g}(x) + (ip - x)\dot{g}(x) - \frac{ip}{2}\left(1 + \frac{a_1}{\mu}\right)g(x) = 0. \tag{56}$$

This is the differential equation for confluent hypergeometric functions. Specifically, the Kummer function $M(a,b,z) = {}_1F_1(a;b;z)$ and the Tricomi hypergeometric function $U(a,b,z)$ are two linearly independent solutions of the equation:

$$z\ddot{w}(z) + (b - z)\dot{w}(z) - aw(z) = 0, \qquad w(z) = M(a,b,z), U(a,b,z). \tag{57}$$

This allows us to write the general solution of Eq. (54) as

$$\psi_2(t) = C_1 e^{-ia_1 t} U\left(\frac{ip}{2}\left(1 + \frac{a_1}{\mu}\right), ip, i\mu t\right) + C_2 e^{-ia_1 t} M\left(\frac{ip}{2}\left(1 + \frac{a_1}{\mu}\right), ip, i\mu t\right). \tag{58}$$

Setting $a_1 = 0$ and using the identities (10.2.3), (10.4.3), (10.16.5), and (10.16.6) in [52], we cast this solution into the form:

$$\psi_2(t)_{(a_1=0)} = (a_2 t)^{-\frac{ip}{2} + \frac{1}{2}}\left(C_1 J_{\frac{ip}{2} - \frac{1}{2}}(a_2 t) + C_2 J_{-\frac{ip}{2} + \frac{1}{2}}(a_2 t)\right), \tag{59}$$

where $J_\alpha(t)$ is the Bessel function of the first kind. Determining the component $\psi_1(t)$ only involves differentiating $\psi_2(t)$ for the 2 × 2 problem. To connect the solutions obtained with the contour integral procedure, we make the following substitution into (51) to recover (14c):

$$\left\{p = -\frac{1}{v},\ a_1 = 0,\ a_2 = -\Delta\right\}. \tag{60}$$

We have $|\psi(t)\rangle = \psi_1(t)|1\rangle + \psi_2(t)|2\rangle$, where $|1\rangle$ and $|2\rangle$ are the ground state is and the excited state at $t = 0^+$, respectively. The solution of the non-stationary Schrödinger equation (51) that starts in the ground state at $t = 0^+$ with parameters (60) is,

$$|\psi_{1/2,0}^1(\tau)\rangle = \left(\frac{\pi}{2\cosh\frac{\pi}{2v}}\right)^{1/2}\tau^{\frac{1}{2} + \frac{i}{2v}}\left[J_{\frac{1}{2} + \frac{i}{2v}}(\tau)|2\rangle - iJ_{-\frac{1}{2} + \frac{i}{2v}}(\tau)|1\rangle\right], \tag{61}$$

where $\tau = t\Delta$. Solving for the wavefunction starting from the excited state $|2\rangle$, we find

$$|\psi_{1/2,0}^2(\tau)\rangle = \left(\frac{\pi}{2\cosh\frac{\pi}{2v}}\right)^{1/2}\tau^{\frac{1}{2} + \frac{i}{2v}}\left[J_{-\frac{1}{2} - \frac{i}{2v}}(\tau)|2\rangle + iJ_{\frac{1}{2} - \frac{i}{2v}}(\tau)|1\rangle\right]. \tag{62}$$

Up to normalization, the ground state solution (61) matches the contour integral result (33), and similarly the excited state solution (62) corresponds to (35).

### 3.1.2 A 3 × 3 HLZ problem

We now turn to the next larger model, Eq. (52), where—analogous to the $2 \times 2$ case—we rewrite the differential equations as follows:

$$t^3 \dddot{\psi}_1(t) + i(p+q)t^2 \ddot{\psi}_1(t) \tag{63a}$$
$$+ \left( -q(p+i) - 2ip + (a_1^2 + a_2^2)t^2 \right) t \dot{\psi}_1(t) + p \left( 2(q+i) + ia_2^2 t^2 \right) \psi_1(t) = 0,$$

$$\psi_2(t) = \frac{1}{a_1} \left[ i\dot{\psi}_1(t) - \frac{p}{t} \psi_1(t) \right], \tag{63b}$$

$$\psi_3(t) = \frac{1}{a_1 a_2} \left[ -\ddot{\psi}_1(t) - \frac{i}{t}(p+q)\dot{\psi}_1(t) + \frac{1}{t^2}(p(q+i) - t^2 a_1^2)\psi_1(t) \right]. \tag{63c}$$

Substituting $\psi_1(t) = t^\beta g(t)$ into (63b) and using $x = -(a_1^2 + a_2^2)t^2/4$, we rewrite the main differential equation as follows

$$x^2 \ddot{g}(x) + \frac{1}{2}x(3\beta + ip + iq + 3)\ddot{g}(x) + \frac{1}{4}\left(3\beta^2 + 2i\beta(p+q) - p(q+i) - 4x\right)\dot{g}(x)$$
$$- \frac{\left(\beta a_1^2 + a_2^2(\beta + ip)\right)}{2\left(a_1^2 + a_2^2\right)}g(x) + \frac{(\beta - 2)(\beta + ip)(\beta + iq - 1)}{8x}g(x) = 0. \tag{64}$$

For the choices $\beta = 2, -ip, 1 - iq$, the above differential equation is equivalent to that for the generalised hypergeometric function[4] $_1F_2(a_1; b_1, b_2; z)$,

$$z^2 \dddot{w}(z) + z(b_1 + b_2 + 1)\ddot{w}(z) + (b_1 b_2 - z)\dot{w}(z) - a_1 w(z) = 0, \tag{65}$$

with the solution $w(z) = {}_1F_2(a_1; b_1, b_2; z)$. As we identified three independent solutions through the choice of $\beta$, we have the general solution of (63b) for $\psi_1(t)$,

$$\psi_1(t) = \mathcal{C}_1 t^2 {}_1F_2 \left[ \begin{matrix} 1 + \frac{ipa_2^2}{2(a_1^2+a_2^2)} \\ 2 + \frac{ip}{2}, \frac{3}{2} + \frac{iq}{2} \end{matrix} ; -\frac{(a_1^2+a_2^2)}{4}t^2 \right] + \mathcal{C}_2 t^{-ip} {}_1F_2 \left[ \begin{matrix} -\frac{ip}{2} + \frac{ipa_2^2}{2(a_1^2+a_2^2)} \\ -\frac{ip}{2}, \frac{1}{2} + \frac{i(q-p)}{2} \end{matrix} ; -\frac{(a_1^2+a_2^2)}{4}t^2 \right]$$
$$+ \mathcal{C}_3 t^{1-iq} {}_1F_2 \left[ \begin{matrix} \frac{1}{2} - \frac{iq}{2} + \frac{ipa_2^2}{2(a_1^2+a_2^2)} \\ \frac{1}{2} - \frac{iq}{2}, \frac{3}{2} + \frac{i(p-q)}{2} \end{matrix} ; -\frac{(a_1^2+a_2^2)}{4}t^2 \right]. \tag{66}$$

To display the full solution, including the components $\psi_2$ and $\psi_3$, we first use the following substitution in (52) to recover (17d):

$$\left\{ p = -\frac{3}{\nu}, q = -\frac{1}{\nu}, a_1 = -\frac{2\Delta}{\sqrt{3}}, a_2 = 2\sqrt{\frac{2}{3}}\Delta \right\}. \tag{67}$$

The solution of the non-stationary Schrödinger equation for the Hamiltonian (17d) starting from the ground state at $t = 0^+$ is

$$|\psi_{1,0}^1(\tau)\rangle = \phi_1^1(\tau)|1\rangle + \phi_2^1(\tau)|2\rangle + \phi_3^1(\tau)|3\rangle, \tag{68}$$

---

[4]Not to be confused with the 'general hypergeometric functions' or 'Gelfand-Aomoto hypergeometric functions' which are also ubiquitous in the literature on KZ equations [34].

where $\tau = t\Delta$ and

$$\phi_1^1(\tau) = e^{\frac{3i}{\nu}\ln(\tau)} {}_1F_2\left[\begin{matrix} \frac{i}{2\nu} \\ \frac{1}{2}+\frac{i}{\nu},\frac{3i}{2\nu} \end{matrix} ; -\tau^2\right], \tag{69a}$$

$$\phi_2^1(\tau) = \frac{2i\,\nu\tau e^{\frac{3i}{\nu}\ln(\tau)}}{(2i+\nu)\sqrt{3}} {}_1F_2\left[\begin{matrix} 1+\frac{i}{2\nu} \\ \frac{3}{2}+\frac{i}{\nu},1+\frac{3i}{2\nu} \end{matrix} ; -\tau^2\right], \tag{69b}$$

$$\phi_3^1(\tau) = \frac{e^{\frac{3i}{\nu}\ln(\tau)}}{\sqrt{2}}\left({}_1F_2\left[\begin{matrix} \frac{i}{2\nu} \\ \frac{1}{2}+\frac{i}{\nu},\frac{3i}{2\nu} \end{matrix} ; -\tau^2\right] - {}_1F_2\left[\begin{matrix} 1+\frac{i}{2\nu} \\ \frac{3}{2}+\frac{i}{\nu},1+\frac{3i}{2\nu} \end{matrix} ; -\tau^2\right]\right) \tag{69c}$$

$$+ \frac{2\sqrt{2}\nu^2(2\nu+i)\tau^2 e^{\frac{3i}{\nu}\ln(\tau)}}{(2\nu+3i)(\nu+2i)(3\nu+2i)} {}_1F_2\left[\begin{matrix} 2+\frac{i}{2\nu} \\ \frac{5}{2}+\frac{i}{\nu},2+\frac{3i}{2\nu} \end{matrix} ; -\tau^2\right].$$

The solutions to the differential equations (51) and (52) can be obtained for arbitrary initial conditions. To determine all transition probabilities, it is necessary to also consider time evolution starting from the excited states. For the $3 \times 3$ case, the wavefunctions corresponding to initial states $|2\rangle$ and $|3\rangle$ are provided in Appendix B.1.

## 3.2 Transition probabilities

### 3.2.1 The $2 \times 2$ HLZ problem

At $t = 0^+$, the ground state of the $2 \times 2$ HLZ model is $|1/2, 0\rangle_g$ in (28). At $t \to \infty$, it becomes $|\uparrow\downarrow\rangle$. The large-time asymptotes of the solutions (61) and (62) read

$$|\psi_{1/2,0}^1(\tau)\rangle \to -i\left(\frac{1}{2\cosh\frac{\pi}{2\nu}}\right)^{1/2}\exp\left(\frac{i\ln\tau}{2\nu}\right)\left(e^{i\tau}e^{\frac{\pi}{4\nu}}|\uparrow\downarrow\rangle + e^{-i\tau}e^{-\frac{\pi}{4\nu}}|\downarrow\uparrow\rangle\right), \tag{70a}$$

$$|\psi_{1/2,0}^2(\tau)\rangle \to -\left(\frac{1}{2\cosh\frac{\pi}{2\nu}}\right)^{1/2}\exp\left(\frac{i\ln\tau}{2\nu}\right)\left(e^{i\tau}e^{-\frac{\pi}{4\nu}}|\uparrow\downarrow\rangle - e^{-i\tau}e^{\frac{\pi}{4\nu}}|\downarrow\uparrow\rangle\right). \tag{70b}$$

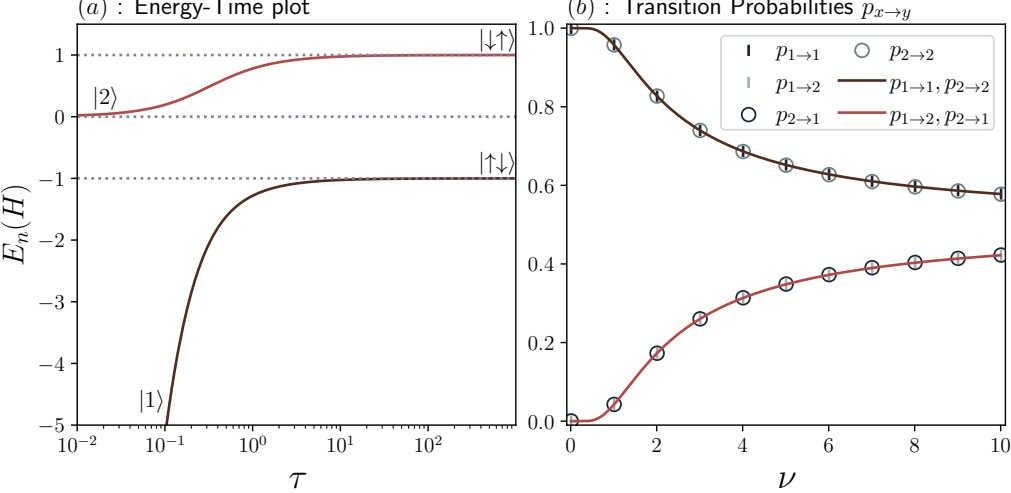

Figure 3: (a) Instantaneous (adiabatic) eigenvalues of the $2 \times 2$ HLZ model as functions of $\tau = t\Delta$ for $\varepsilon_1 = 1$, $\varepsilon_2 = 2$ and $\nu = 2$. The ground state $|1\rangle$ evolves towards $|\uparrow\downarrow\rangle$. The excited state $|2\rangle$ ends up in $|\downarrow\uparrow\rangle$. (b) Elements of the transition probability matrix. Solid lines represent the analytical expressions in (71). The scatter plots represent the numerical simulation of $p_{x\to y}$ at $\tau = 10^3$.

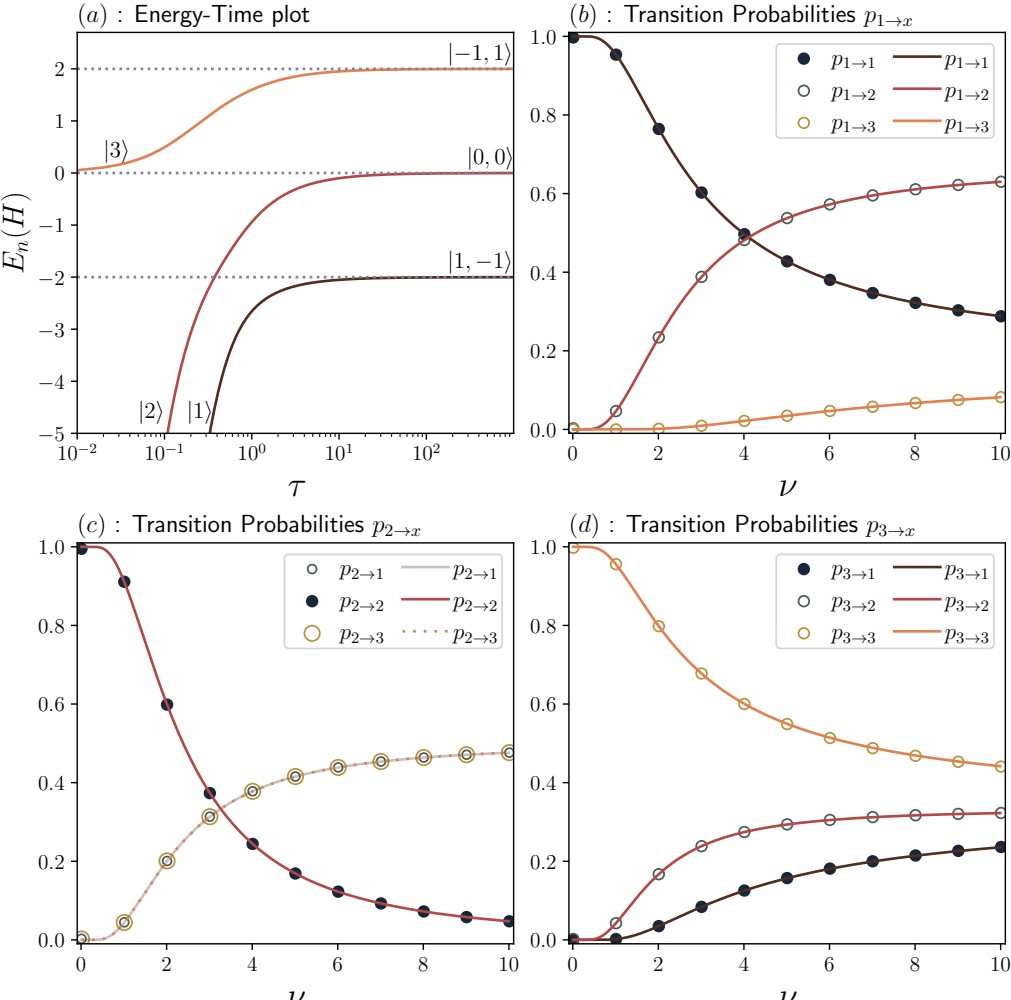

Figure 4: (a) Instantaneous eigenvalues vs $\tau = t\Delta$ for the $3 \times 3$ HLZ model with parameters (67) ($\varepsilon_1 = 1$, $\varepsilon_2 = 2$ and $v = 2$). The ground state $|1\rangle$ evolves towards $|1,-1\rangle$. $|2\rangle$ and $|3\rangle$ evolve to $|0,0\rangle$ and $|-1,1\rangle$, respectively. (b–d) Transition probabilities. The solid lines represent the analytical expressions in (73). The scatter plots are the numerical simulation of $p_{x \to y}$ for different values of $v$ evaluated at $\tau = 10^3$.

This implies

$$P_{2\times 2} = \frac{1}{e^{\frac{\pi}{2v}} + e^{-\frac{\pi}{2v}}} \begin{pmatrix} e^{\frac{\pi}{2v}} & e^{-\frac{\pi}{2v}} \\ e^{-\frac{\pi}{2v}} & e^{\frac{\pi}{2v}} \end{pmatrix}. \tag{71}$$

Fig. 3 shows these transition probabilities, alongside the spectrum of the $2 \times 2$ model for various choices of $v$ at large $\tau = \Delta t$.

### 3.2.2 The $3 \times 3$ HLZ problem

The instantaneous ground state of the $3 \times 3$ HLZ model at $t = 0^+$ is $|1,0\rangle_g$ in (18). At $t \to \infty$, it becomes $|1,-1\rangle$. Evaluating the asymptotic behaviour of the solutions (68) and (B.1) at large

time, we find

$$|\psi^1_{1,0}(\tau)\rangle \to C_1 \tau^{\frac{2i}{\nu}} \left[ e^{\frac{\pi}{\nu}+2i\tau-\frac{i}{\nu}\ln\tau} |1,\text{-}1\rangle + \frac{2\pi e^{-\frac{i}{\nu}\ln 2}}{\Gamma\left(\frac{i}{2\nu}+\frac{1}{2}\right)^2} |0,0\rangle + e^{-\left(\frac{\pi}{\nu}+2i\tau+\frac{i}{\nu}\ln\tau\right)} |\text{-}1,1\rangle \right], \quad (72a)$$

$$|\psi^2_{1,0}(\tau)\rangle \to C_2 \tau^{\frac{2i}{\nu}} \sinh\left(\frac{\pi}{2\nu}\right) \left[ e^{-\frac{i}{\nu}\ln\tau} (|1,\text{-}1\rangle - |\text{-}1,1\rangle) + \frac{i\sqrt{\pi}\Gamma\left(\frac{1}{2}-\frac{i}{2\nu}\right)}{\Gamma\left(1-\frac{i}{2\nu}\right)\Gamma\left(\frac{i}{\nu}\right)} |0,0\rangle \right], \quad (72b)$$

$$|\psi^3_{1,0}(\tau)\rangle \to C_3 \tau^{\frac{2i}{\nu}} \left[ \frac{e^{2i\tau-\frac{i}{\nu}\ln\tau}}{1+e^{\frac{\pi}{\nu}}} |1,\text{-}1\rangle - \frac{e^{-\frac{i}{\nu}\ln 2}\Gamma\left(\frac{-i+\nu}{2\nu}\right)}{2\Gamma\left(\frac{i+\nu}{2\nu}\right)\cosh\left(\frac{\pi}{2\nu}\right)} |0,0\rangle + \frac{e^{\frac{\pi}{\nu}-2i\tau-\frac{i}{\nu}\ln\tau}}{1+e^{\frac{\pi}{\nu}}} |\text{-}1,1\rangle \right]. \quad (72c)$$

Here,

$$C_1 = \frac{\sqrt{\frac{3}{2\pi}}\Gamma\left(\frac{1}{2}+\frac{i}{\nu}\right)\Gamma\left(\frac{3i}{2\nu}\right)}{\Gamma\left(\frac{i}{2\nu}\right)}, \qquad C_2 = \mathcal{N}_2 \sqrt{\frac{3}{2}} e^{\frac{2i}{\nu}\ln 2} \frac{\Gamma\left(2-\frac{2i}{\nu}\right)\Gamma\left(\frac{i}{\nu}\right)}{\Gamma\left(\frac{1}{2}-\frac{i}{2\nu}\right)^2}, \quad (72d)$$

and

$$C_3 = \mathcal{N}_3 \frac{\sqrt{6\pi}\Gamma\left(2-\frac{3i}{2\nu}\right)}{\Gamma\left(-\frac{1}{2}+\frac{i}{2\nu}\right)\Gamma\left(1-\frac{i}{\nu}\right)}. \quad (72e)$$

The transition probability matrix is

$$P_{3\times 3} = \begin{pmatrix} \frac{1}{\left(1+e^{-\frac{2\pi}{\nu}}\right)\left(1+e^{-\frac{\pi}{\nu}}+e^{-\frac{2\pi}{\nu}}\right)} & \frac{e^{-\frac{\pi}{\nu}}\left(1+e^{-\frac{\pi}{\nu}}\right)^2}{\left(1+e^{-\frac{2\pi}{\nu}}\right)\left(1+e^{-\frac{\pi}{\nu}}+e^{-\frac{2\pi}{\nu}}\right)} & \frac{e^{-\frac{4\pi}{\nu}}}{\left(1+e^{-\frac{2\pi}{\nu}}\right)\left(1+e^{-\frac{\pi}{\nu}}+e^{-\frac{2\pi}{\nu}}\right)} \\ \frac{1}{2\cosh\left(\frac{\pi}{\nu}\right)} & 1-\frac{1}{\cosh\left(\frac{\pi}{\nu}\right)} & \frac{1}{2\cosh\left(\frac{\pi}{\nu}\right)} \\ \frac{1}{1+e^{\frac{\pi}{\nu}}+e^{\frac{2\pi}{\nu}}} & \frac{1}{2\cosh\left(\frac{\pi}{\nu}\right)+1} & \frac{1}{1+e^{-\frac{\pi}{\nu}}+e^{-\frac{2\pi}{\nu}}} \end{pmatrix}. \quad (73)$$

A plot of the transition probabilities, alongside the adiabatic spectrum of the $3 \times 3$ HLZ model for various choices of $\nu$ at large $\tau = \Delta t$ is provided in Figure 4.

Thus, we conclude our brute force investigation of the two hyperbolic Landau-Zener problems. As mentioned earlier, the transition probabilities derived in this subsection have also been found through other means by Sinitsyn [45], who used symmetries and the no-go constraints for solving the transition probability matrix. For example, for the $2 \times 2$ model, the second row in the transition probability matrix (71) is straightforwardly determined by time reversal symmetry. However, for larger (H)LZ problems (i.e. $4 \times 4$ or larger), we believe that all symmetries and constraints do not provide enough information to find the transition probabilities, requiring analytical solutions to the Schrödinger equation after all. Nonetheless, for the models described in this section a full solution in terms of wavefunctions is now also available.

# 4 New $3 \times 3$ and $4 \times 4$ integrable HLZ models

With the connection between the KZ equations and HLZ models established, we now proceed to discuss three more examples of HLZ models that derive from the time-dependent BCS Hamiltonian (9). We will also make more general statements on higher order HLZ problems. All these models are integrable, since they are derived from the KZ equations. However, we determine solutions in terms of known functions only for some of them.

## 4.1 Three-site spin-1/2 BCS-derived HLZ problem

We revisit a problem derived from the three-site spin-1/2 BCS Hamiltonian. It is found within the $S^z = -1/2$ sector, see Eq. (38). First we will consider the case where $\varepsilon_2 = \frac{1}{2}(\varepsilon_1 + \varepsilon_3)$,

which after the basis transformation (43), is written as

$$i\begin{pmatrix}\dot{\psi}_1(t)\\\dot{\psi}_2(t)\\\dot{\psi}_3(t)\end{pmatrix}=\begin{pmatrix}-\frac{3}{2\nu t}&\sqrt{\frac{2}{3}}\Delta&0\\\sqrt{\frac{2}{3}}\Delta&0&-\frac{\Delta}{\sqrt{3}}\\0&-\frac{\Delta}{\sqrt{3}}&0\end{pmatrix}\begin{pmatrix}\psi_1(t)\\\psi_2(t)\\\psi_3(t)\end{pmatrix},\tag{74}$$

where $\Delta=\frac{1}{2}(\varepsilon_3-\varepsilon_1)$. The above model is of the form of (52) with parameters given by:

$$\left\{p=-\frac{3}{2\nu},\,q=0,\,a_1=\sqrt{\frac{2}{3}}\Delta,\,a_2=-\frac{\Delta}{\sqrt{3}}\right\}.\tag{75}$$

This model is solved in its entirety in terms of the $_1F_2$ hypergeometric functions in the previous section. For arbitrary choices of $\varepsilon$, the ground-state initial condition solution is available in terms of the Kampé de Fériet functions, as described by the contour integral solution in Sec. 2.1.2. The probability transition matrix is calculated to be

$$P_{3\times3}=\begin{pmatrix}\frac{e^{\frac{\pi}{\nu}}}{1+2\cosh\left(\frac{\pi}{\nu}\right)}&\frac{1}{1+2\cosh\left(\frac{\pi}{\nu}\right)}&\frac{1}{1+e^{\frac{\pi}{\nu}}+e^{\frac{2\pi}{\nu}}}\\\frac{1}{2\left(1+e^{\frac{\pi}{\nu}}-e^{\frac{\pi}{2\nu}}\right)}&\frac{\left(e^{\frac{\pi}{2\nu}}-1\right)^2}{2\left(1+e^{\frac{\pi}{\nu}}-e^{\frac{\pi}{2\nu}}\right)}&\frac{e^{\frac{\pi}{\nu}}}{2\left(1+e^{\frac{\pi}{\nu}}-e^{\frac{\pi}{2\nu}}\right)}\\\frac{1}{2\left(1+e^{\frac{\pi}{\nu}}+e^{\frac{\pi}{2\nu}}\right)}&\frac{1}{2}+\frac{1}{2\left(1+2\cosh\left(\frac{\pi}{2\nu}\right)\right)}&\frac{e^{\frac{\pi}{\nu}}}{2\left(1+e^{\frac{\pi}{\nu}}+e^{\frac{\pi}{2\nu}}\right)}\end{pmatrix}.\tag{76}$$

A plot of the transition probabilities and the energy spectrum as a function of time is provided in Figure 5.

Another choice of $\varepsilon$ for which we study the three level system (38) is $\varepsilon_i=\varepsilon_j$ for some $i\neq j$ with $\varepsilon_1\leq\varepsilon_2\leq\varepsilon_3$. In this case, the middle energy level coalesces with either the ground state ($\varepsilon_1=\varepsilon_2$) or the highest level ($\varepsilon_2=\varepsilon_3$). Here we consider $\varepsilon_2=\varepsilon_3$. Then after rotation to the following basis:

$$\begin{aligned}|1\rangle&=\frac{1}{\sqrt{3}}\left(|\uparrow\downarrow\downarrow\rangle+|\downarrow\uparrow\downarrow\rangle+|\downarrow\downarrow\uparrow\rangle\right),\\|2\rangle&=\frac{1}{\sqrt{6}}\left(-2|\uparrow\downarrow\downarrow\rangle+|\downarrow\uparrow\downarrow\rangle+|\downarrow\downarrow\uparrow\rangle\right),\\|3\rangle&=-\frac{1}{\sqrt{2}}\left(|\downarrow\uparrow\downarrow\rangle-|\downarrow\downarrow\uparrow\rangle\right),\end{aligned}\tag{77}$$

and setting $\varepsilon_2-\varepsilon_1=\Delta$, the model takes the form:

$$i\begin{pmatrix}\dot{\psi}_1(t)\\\dot{\psi}_2(t)\\\dot{\psi}_3(t)\end{pmatrix}=\left[\begin{pmatrix}\frac{2\Delta}{3}-\frac{3}{2\nu t}&\frac{2\sqrt{2}\Delta}{3}&0\\\frac{2\sqrt{2}\Delta}{3}&0&0\\0&0&\frac{4\Delta}{3}\end{pmatrix}-\left(\frac{4\Delta}{3}+\varepsilon_1\right)\mathbb{I}\right]\begin{pmatrix}\psi_1(t)\\\psi_2(t)\\\psi_3(t)\end{pmatrix}.\tag{78}$$

We note that the problem reduces to solving the general $2\times2$ HLZ problem (51), whose solutions are available in terms of confluent hypergeometric functions. Equivalently, one may understand this as a mixed-spin problem as investigated in appendix C. The probability transition matrix for this problem is given by:

$$P_{3\times3}=\begin{pmatrix}\frac{e^{\frac{2\pi}{\nu}}}{1+e^{\frac{\pi}{\nu}}+e^{\frac{2\pi}{\nu}}}&\frac{1+e^{\frac{\pi}{\nu}}}{2\left(1+e^{\frac{\pi}{\nu}}+e^{\frac{2\pi}{\nu}}\right)}&\frac{1+e^{\frac{\pi}{\nu}}}{2\left(1+e^{\frac{\pi}{\nu}}+e^{\frac{2\pi}{\nu}}\right)}\\\frac{1+e^{\frac{\pi}{\nu}}}{1+e^{\frac{\pi}{\nu}}+e^{\frac{2\pi}{\nu}}}&\frac{e^{\frac{2\pi}{\nu}}}{2\left(1+e^{\frac{\pi}{\nu}}+e^{\frac{2\pi}{\nu}}\right)}&\frac{e^{\frac{2\pi}{\nu}}}{2\left(1+e^{\frac{\pi}{\nu}}+e^{\frac{2\pi}{\nu}}\right)}\\0&\frac{1}{2}&\frac{1}{2}\end{pmatrix}.\tag{79}$$

A plot of the transition probabilities and the energy spectrum as a function of time is provided in Figure 6.

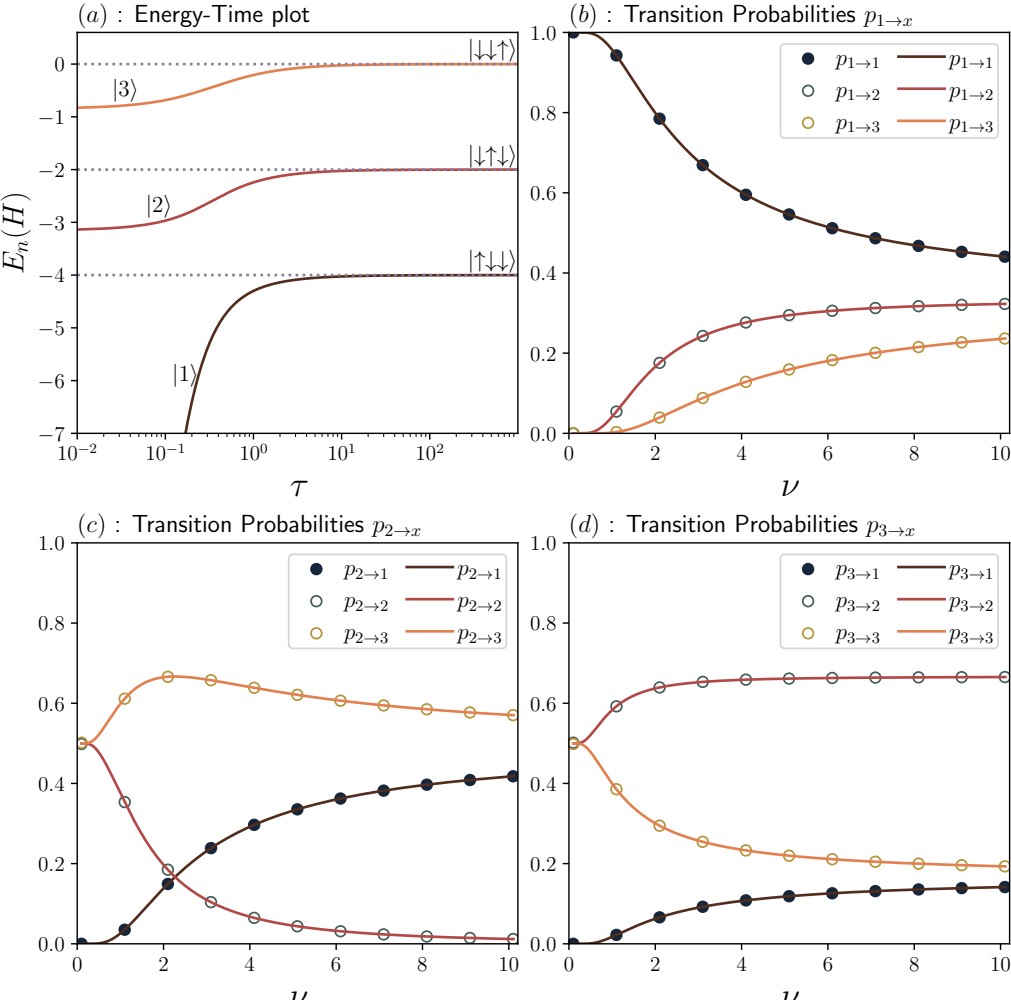

Figure 5: (a) Instantaneous eigenvalues of the $3 \times 3$ HLZ model (74) with $\varepsilon_1 = 1$, $\varepsilon_2 = 2$, $\varepsilon_3 = 3$ and $\tau = t\Delta$ for $\nu = 2$. The ground state $|1\rangle$ evolves towards $|\uparrow\downarrow\downarrow\rangle$. $|2\rangle$ and $|3\rangle$ evolve to $|\downarrow\uparrow\downarrow\rangle$ and $|\downarrow\downarrow\uparrow\rangle$ respectively. (b–d) Elements of the transition probability matrix, where the solid lines represent the analytical expressions in (76). The scatter plots represent the numerical simulation of $p_{x \to y}$ as a function of $\nu$ at $\tau = 10^3$.

## 4.2 Four-site spin-1/2 BCS-derived HLZ problem

Similarly to the previous section, we now consider a spin-1/2 BCS Hamiltonian but with four sites, and focus on the $S^z = -1$ sector,

$$
H_{1/2}^{(-4 \times 1/2 + 1)} = \sum_{i=1}^{4} \varepsilon_i \mathbb{I}_{4 \times 4} - 2 \begin{pmatrix} \varepsilon_1 & 0 & 0 & 0 \\ 0 & \varepsilon_2 & 0 & 0 \\ 0 & 0 & \varepsilon_3 & 0 \\ 0 & 0 & 0 & \varepsilon_4 \end{pmatrix} - \frac{1}{2\nu t} \begin{pmatrix} 1 & 1 & 1 & 1 \\ 1 & 1 & 1 & 1 \\ 1 & 1 & 1 & 1 \\ 1 & 1 & 1 & 1 \end{pmatrix} . \tag{80}
$$

The solution starting from the ground state is available in terms of three-variable hypergeometric function through evaluating the corresponding contour integral, see Appendix A.3.

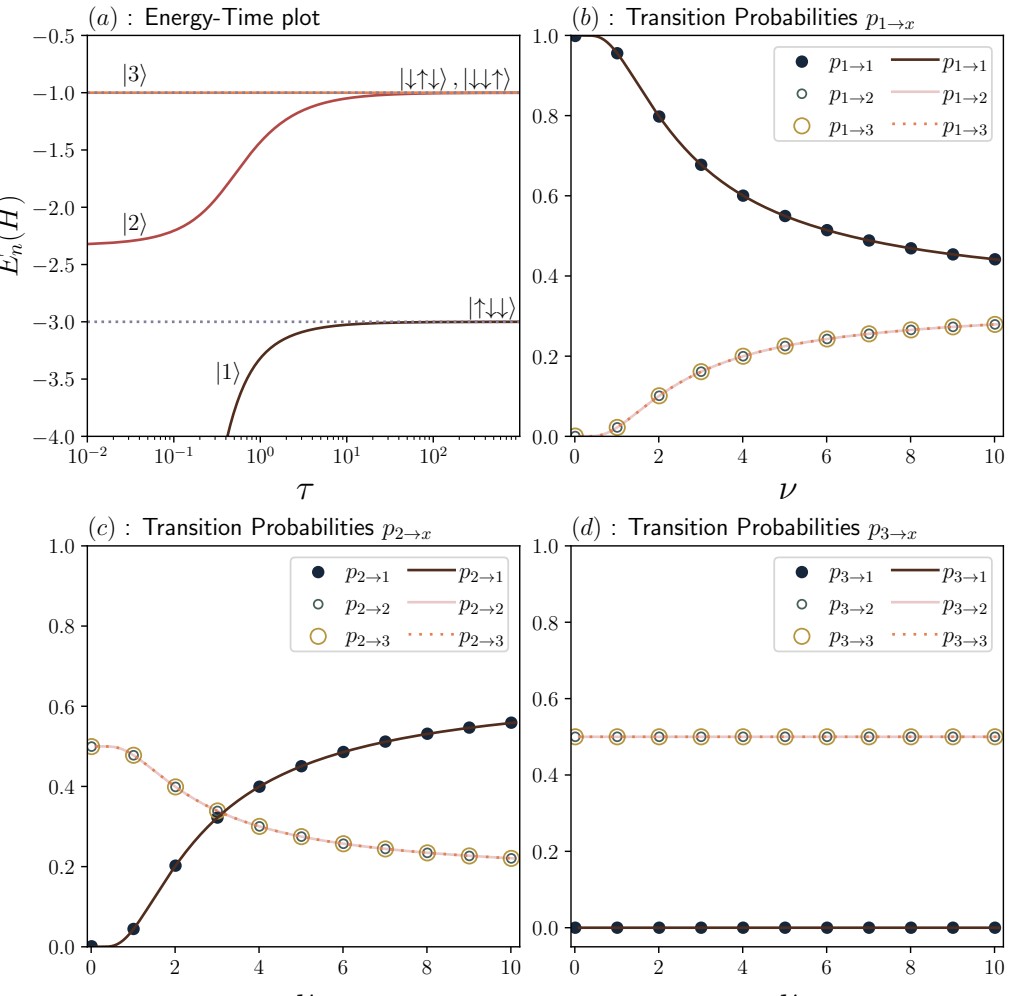

Figure 6: (a) Instantaneous eigenvalues of the $3 \times 3$ HLZ model (78) vs $\tau = t\Delta$ for $\varepsilon_1 = 1$, $\varepsilon_2 = \varepsilon_3 = 2$ and $\nu = 2$. The ground state $|1\rangle$ evolves towards $|\uparrow\downarrow\downarrow\rangle$. $|2\rangle$ and $|3\rangle$ evolve to $|\downarrow\uparrow\downarrow\rangle$ and $|\downarrow\downarrow\uparrow\rangle$ respectively. (b–d) Elements of the transition probability matrix, where the solid lines represent the analytical expressions in (79). The scatter plots represent the numerical simulation of $p_{x \to y}$ as a function of $\nu$ at $\tau = 10^3$.

In this section, we study the case $\varepsilon_1 \leq \varepsilon_2 = \varepsilon_3 \leq \varepsilon_4$. With this choice of $\varepsilon$, the problem becomes tractable analytically, using results from Sec. 3.1. Rotating (80) to the following basis:

$$
\begin{aligned}
|1\rangle &= \frac{1}{2}|\uparrow\downarrow\downarrow\downarrow\rangle + \frac{1}{2}|\downarrow\uparrow\downarrow\downarrow\rangle + \frac{1}{2}|\downarrow\downarrow\uparrow\downarrow\rangle + \frac{1}{2}|\downarrow\downarrow\downarrow\uparrow\rangle \, , \\
|2\rangle &= \frac{1}{\sqrt{2}}|\uparrow\downarrow\downarrow\downarrow\rangle - \frac{1}{\sqrt{2}}|\downarrow\downarrow\downarrow\uparrow\rangle \, , \\
|3\rangle &= \frac{1}{2}|\uparrow\downarrow\downarrow\downarrow\rangle - \frac{1}{2}|\downarrow\uparrow\downarrow\downarrow\rangle - \frac{1}{2}|\downarrow\downarrow\uparrow\downarrow\rangle + \frac{1}{2}|\downarrow\downarrow\downarrow\uparrow\rangle \, , \\
|4\rangle &= -\frac{1}{\sqrt{2}}|\downarrow\uparrow\downarrow\downarrow\rangle + \frac{1}{\sqrt{2}}|\downarrow\downarrow\uparrow\downarrow\rangle \, ,
\end{aligned}
\tag{81}
$$

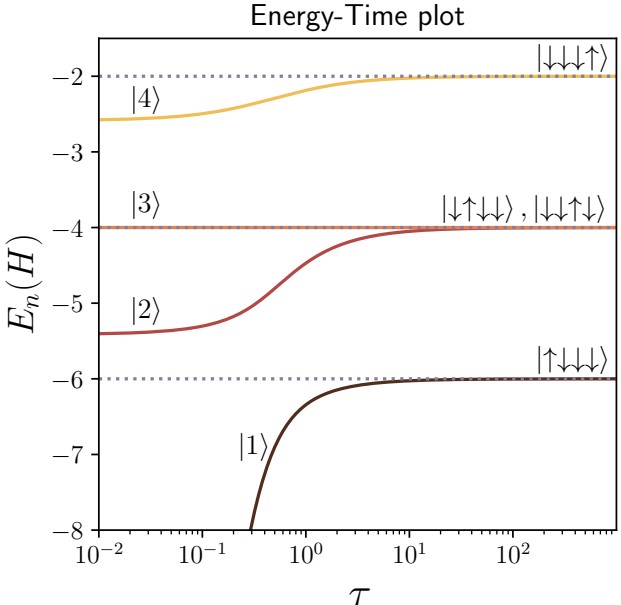

Figure 7: Instantaneous eigenvalues of $H_{1/2}^{(-1)}$ in (82) as functions of time $\tau = t\Delta$ for $\varepsilon_1 = 1$, $\varepsilon_2 = 2$, and $\nu = 2$. The BCS ground state $|1\rangle$ evolves towards $|\uparrow\downarrow\downarrow\rangle$ which represents the lowest energy curve.

and setting $\varepsilon_3 = \varepsilon_2$, $\varepsilon_2 = \Delta + \varepsilon_1$ and $\varepsilon_4 = 2\Delta + \varepsilon_1$, we block diagonalise the Hamiltonian,

$$
i\begin{pmatrix} \dot{\psi}_1(t) \\ \dot{\psi}_2(t) \\ \dot{\psi}_3(t) \\ \dot{\psi}_4(t) \end{pmatrix} = \left[ \begin{pmatrix} -\frac{2}{\nu t} & -\sqrt{2}\Delta & 0 & 0 \\ -\sqrt{2}\Delta & 0 & -\sqrt{2}\Delta & 0 \\ 0 & -\sqrt{2}\Delta & 0 & 0 \\ 0 & 0 & 0 & 0 \end{pmatrix} - 2(\Delta + \varepsilon_1)\mathbb{I}_{4\times4} \right] \begin{pmatrix} \psi_1(t) \\ \psi_2(t) \\ \psi_3(t) \\ \psi_4(t) \end{pmatrix}. \tag{82}
$$

The plot of energy spectrum of the model as a function of time is given in Fig. 7. The problem thus reduces to the $3 \times 3$ model of the form (52), which makes this case completely solvable.

At the same time, as mentioned before, the time evolution with the general Hamiltonian (80) starting from the ground state is accessible through contour integrals (see Appendix A.3). The probability transition matrix for the HLZ problem defined by Eq. (82) is:

$$
P_{4\times4} = \begin{pmatrix} \frac{e^{\frac{3\pi}{\nu}}}{\left(1+e^{\frac{\pi}{\nu}}\right)\left(1+e^{\frac{2\pi}{\nu}}\right)} & \frac{e^{\frac{\pi}{\nu}}}{2\left(1+e^{\frac{2\pi}{\nu}}\right)} & \frac{e^{\frac{\pi}{\nu}}}{2\left(1+e^{\frac{2\pi}{\nu}}\right)} & \frac{1}{\left(1+e^{\frac{\pi}{\nu}}\right)\left(1+e^{\frac{2\pi}{\nu}}\right)} \\ \frac{1+e^{\frac{\pi}{\nu}}}{2\left(1+e^{\frac{2\pi}{\nu}}\right)} & \frac{\left(e^{\frac{\pi}{\nu}}-1\right)^2}{4\left(1+e^{\frac{2\pi}{\nu}}\right)} & \frac{\left(e^{\frac{\pi}{\nu}}-1\right)^2}{4\left(1+e^{\frac{2\pi}{\nu}}\right)} & \frac{e^{\frac{\pi}{\nu}}+e^{\frac{2\pi}{\nu}}}{2\left(1+e^{\frac{2\pi}{\nu}}\right)} \\ \frac{1}{2\left(1+e^{\frac{\pi}{\nu}}\right)} & \frac{1}{4} & \frac{1}{4} & \frac{e^{\frac{\pi}{\nu}}}{2\left(1+e^{\frac{\pi}{\nu}}\right)} \\ \frac{1}{2} & 0 & \frac{1}{2} & 0 \end{pmatrix}. \tag{83}
$$

A plot of the transition probabilities is provided in Figure 8.

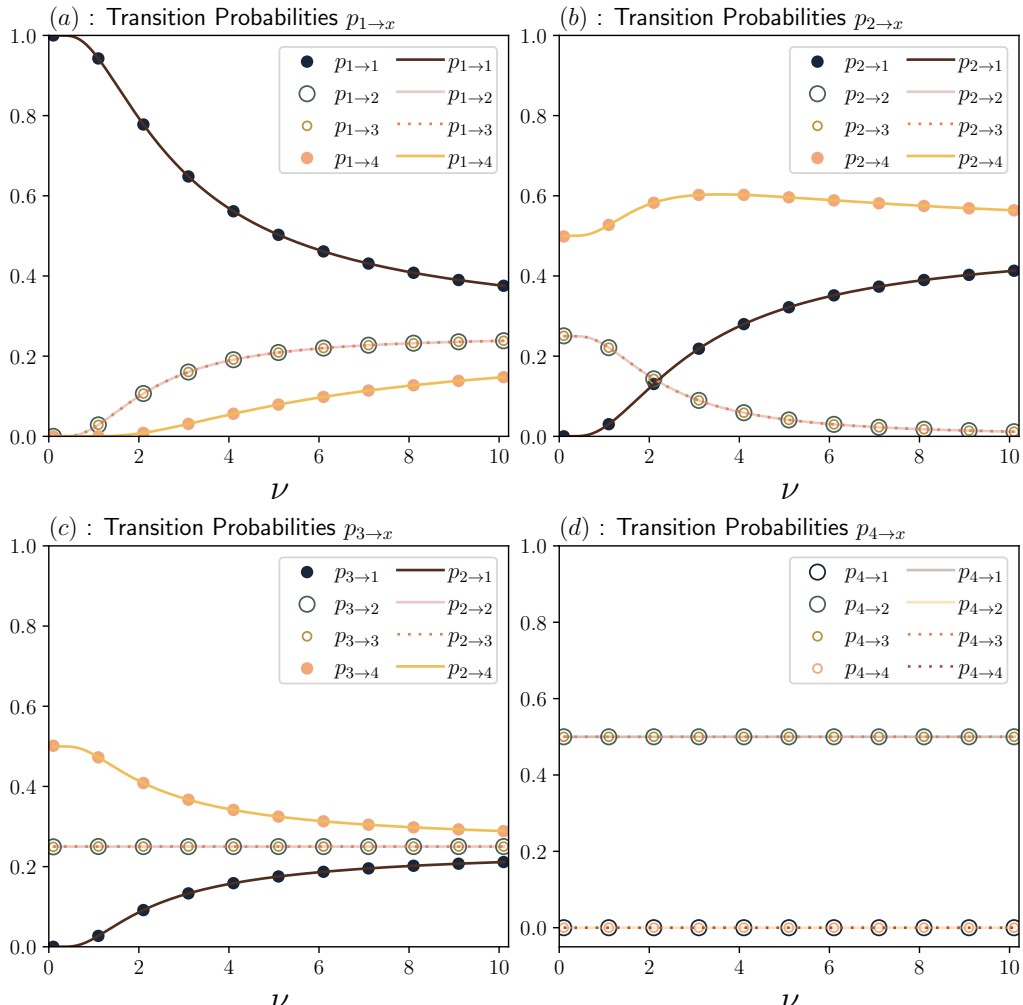

Figure 8: Elements of the transition probability matrix, where the solid lines represent the analytical expressions in (83). The scatter plots represent the numerical simulation of $p_{x\to y}$ as a function of $\nu$ at $\tau = 10^3$.

## 4.3 Two-site spin-3/2 BCS-derived HLZ problem

Here we consider a HLZ problem that is derived from a spin-3/2 BCS Hamiltonian. We introduce the spin-3/2 operators,

$$\hat{s}^z_{3/2} = \frac{1}{2}\begin{pmatrix} 3 & 0 & 0 & 0 \\ 0 & 1 & 0 & 0 \\ 0 & 0 & -1 & 0 \\ 0 & 0 & 0 & -3 \end{pmatrix}, \qquad \hat{s}^+_{3/2} = \begin{pmatrix} 0 & \sqrt{3} & 0 & 0 \\ 0 & 0 & 2 & 0 \\ 0 & 0 & 0 & \sqrt{3} \\ 0 & 0 & 0 & 0 \end{pmatrix}, \qquad \hat{s}^-_{3/2} = \begin{pmatrix} 0 & 0 & 0 & 0 \\ \sqrt{3} & 0 & 0 & 0 \\ 0 & 2 & 0 & 0 \\ 0 & 0 & \sqrt{3} & 0 \end{pmatrix}. \quad (84)$$

We write the wavefunctions in terms of the basis states $|i\rangle$, $i = \pm 1, \pm 3$ as in (12) where $|i\rangle$ is the eigenstate of $\hat{s}^z_{3/2}$ with eigenvalue $i/2$. By ordering the $(i, j)$ indices as

$$
\begin{aligned}
[(-3,-3), \\
(-1,-3),(-3,-1), \\
(1,-3),(-1,-1),(-3,1), \\
(3,-3),(1,-1),(-1,1),(-3,3), \\
(3,-1),(1,1),(-1,3), \\
(3,1),(1,3), \\
(3,3)],
\end{aligned}
\tag{85}
$$

we find

$$
\hat{H}_{\mathrm{BCS},3/2} = \bigoplus_{S^z=-3}^{3} H^{(S^z)}_{3/2},
\tag{86}
$$

where

$$
\begin{aligned}
H^{(\pm 3)}_{3/2}(\nu) &= 3H^{(\pm 1)}_{1/2}(\nu), \\
H^{(-2)}_{3/2}(\nu) &= H^{(0)}_{1/2}(\nu/3) - 2(\varepsilon_1 + \varepsilon_2)\mathbb{I}, \\
H^{(2)}_{3/2}(\nu) &= H^{(0)}_{1/2}(\nu/3) + 2\left(\varepsilon_1 + \varepsilon_2 - \frac{1}{\nu t}\right)\mathbb{I}, \\
H^{(-1)}_{3/2}(\nu) &= H^{(0)}_{1}(\nu/\sqrt{3}) + (\varepsilon_1 + \varepsilon_2)\mathbb{I} - \frac{1}{\nu t}\mathbf{diag}(2-\sqrt{3}, 3-2\sqrt{3}, 2-\sqrt{3}), \\
H^{(1)}_{3/2}(\nu) &= H^{(0)}_{1}(\nu/\sqrt{3}) + (\varepsilon_1 + \varepsilon_2)\mathbb{I} - \frac{1}{\nu t}\mathbf{diag}(3-\sqrt{3}, 4-2\sqrt{3}, 3-\sqrt{3}).
\end{aligned}
\tag{87}
$$

It is worth noting that we cannot solve HLZ problem for $H^{(\pm 1)}_{3/2}$ through any re-scaling of the $S^z = 0$ solution of the spin-1 case. This is because in this case, the Hamiltonian is not shifted by a term proportional to unity as before, but rather the individual (adiabatic) energy levels have shifted. Since the equations are still fundamentally the same, the solution to this problem is still given in terms of the ${}_1F_2$ hypergeometric functions (see Appendix B.2).

The $S^z = 0$ sector is written using the spin-3/2 (84) operators as

$$
\begin{aligned}
H^{(0)}_{3/2}(\nu) = {}& 2(\varepsilon_1 - \varepsilon_2)\hat{s}^z_{3/2} - \frac{1}{\nu t}\Bigg[\frac{1}{2}\left(\hat{s}^+_{3/2} \cdot \hat{s}^-_{3/2} + \hat{s}^-_{3/2} \cdot \hat{s}^+_{3/2}\right) \\
& + \left(\frac{3}{2} - 2\sqrt{3}\right)\left(\hat{s}^+_{3/2} \cdot \hat{s}^+_{3/2} \cdot \hat{s}^-_{3/2} + \hat{s}^-_{3/2} \cdot \hat{s}^-_{3/2} \cdot \hat{s}^+_{3/2}\right) \\
& + \left(3\sqrt{3} - 2\right)\left(\hat{s}^+_{3/2} \cdot \hat{s}^-_{3/2} \cdot \hat{s}^+_{3/2} + \hat{s}^-_{3/2} \cdot \hat{s}^+_{3/2} \cdot \hat{s}^-_{3/2}\right) \\
& + \left(\frac{3}{2} - 2\sqrt{3}\right)\left(\hat{s}^-_{3/2} \cdot \hat{s}^+_{3/2} \cdot \hat{s}^+_{3/2} + \hat{s}^+_{3/2} \cdot \hat{s}^-_{3/2} \cdot \hat{s}^-_{3/2}\right)\Bigg].
\end{aligned}
\tag{88}
$$

## 4.4 Remarks on $N \geq 4$ models

Here we provide some remarks on the general multi-state problems that can be constructed by going to higher spin-$s$ representations. We do not attempt to derive the general solution of the non-stationary Schrödinger equation for the $H^{(0)}_{3/2}$ block in equation (88), and solve the corresponding $4 \times 4$ HLZ problem numerically.

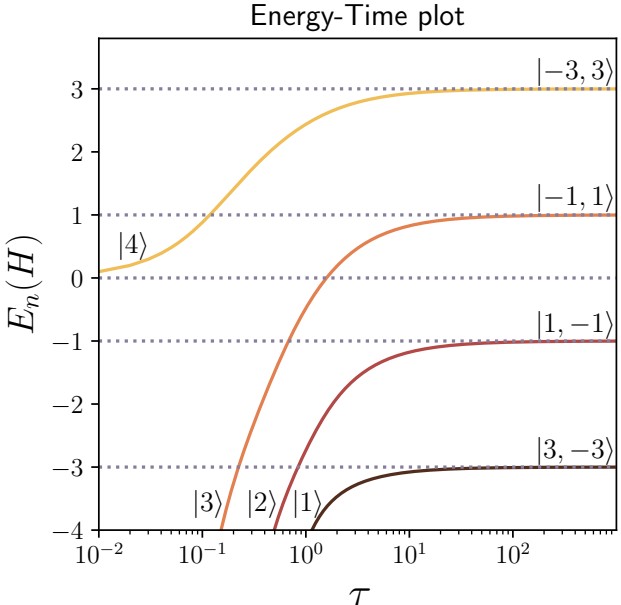

Figure 9: Instantaneous eigenvalues of $H_{3/2}^{(0)}$ as functions of time $\tau = t\Delta$ for $\varepsilon_1 = 1$, $\varepsilon_2 = 2$, and $\nu = 2$. The BCS ground state $|1\rangle$ evolves towards $|3,\text{-}3\rangle$ which represents the lowest energy curve.

First, it is interesting to note that (28) and (18) are also the eigenstates of $\hat{S}^+\hat{S}^- = \sum_{j,k} \hat{s}_j^+ \hat{s}_k^-$ in their corresponding spin representation. By choosing the correct eigenstates of $\hat{S}^+\hat{S}^-$ of a given $S^z$ sector, we can construct the unitary transformation to simplify $H^{(S^z)}$ blocks of arbitrary spin-$s$. For instance, for $S^z = 0$ this allows us to define the following orthogonal transformation:

$$
\begin{aligned}
|3/2, 0\rangle_g \equiv |1\rangle &= \frac{1}{2\sqrt{5}} \left( |3,\text{-}3\rangle + 3|1,\text{-}1\rangle + 3|\text{-}1, 1\rangle + |\text{-}3, 3\rangle \right), \\
|2\rangle &= \frac{1}{2} \left( -|3,\text{-}3\rangle - |1,\text{-}1\rangle + |\text{-}1, 1\rangle + |\text{-}3, 3\rangle \right), \\
|3\rangle &= \frac{1}{2\sqrt{5}} \left( 3|3,\text{-}3\rangle - |1,\text{-}1\rangle - |\text{-}1, 1\rangle + 3|\text{-}3, 3\rangle \right), \\
|4\rangle &= \frac{1}{2} \left( -|3,\text{-}3\rangle + |1,\text{-}1\rangle - |\text{-}1, 1\rangle + |\text{-}3, 3\rangle \right).
\end{aligned}
\tag{89}
$$

Then, the differential equation of $H_{3/2}^{(0)}$ takes the form

$$
i \begin{pmatrix} \dot{\psi}_1(t) \\ \dot{\psi}_2(t) \\ \dot{\psi}_3(t) \\ \dot{\psi}_4(t) \end{pmatrix} = \begin{pmatrix} -\frac{6}{\nu t} & \frac{3\Delta}{\sqrt{5}} & 0 & 0 \\ \frac{3\Delta}{\sqrt{5}} & -\frac{3}{\nu t} & \frac{4\Delta}{\sqrt{5}} & 0 \\ 0 & \frac{4\Delta}{\sqrt{5}} & -\frac{1}{\nu t} & \sqrt{5}\Delta \\ 0 & 0 & \sqrt{5}\Delta & 0 \end{pmatrix} \begin{pmatrix} \psi_1(t) \\ \psi_2(t) \\ \psi_3(t) \\ \psi_4(t) \end{pmatrix},
\tag{90}
$$

where $\Delta = \varepsilon_1 - \varepsilon_2$. All HLZ models in this work are represented by tridiagonal matrices, with only the diagonal elements being time-dependent ($\propto 1/t$) in the diabatic basis. In fact, this is the general appearance of any $N \times N$ representation of our model. The plots of instantaneous (adiabatic) eigenvalues of these models also have a familiar behaviour: at $t = 0^+$, the BCS ground state evolves from the lowest energy, while the highest energy band begins at $E = 0$. At $t \to \infty$, they all tend towards one of the $|i\rangle \otimes |j\rangle$ states, see Figure 9.

Finally, it is always possible to identify the transition probabilities in the limits $v \to 0$ (adiabatic, assuming there are no degeneracy) and $v \to \infty$ (diabatic). In the adiabatic limit, $p_{x \to x} = 1$, while the remaining probabilities are zero. In the diabatic limit, each $p_{x \to y}$ is given by the weighted coefficients of the basis transformation in (89). We numerically validate this in Figure 10 for $s = 3/2$, and it can be further verified with the help of (71) and (73). Additionally, using a saddle-point approach, one can always compute each $p_{1 \to y}$ [5] for arbitrary system size, as was done in [36].

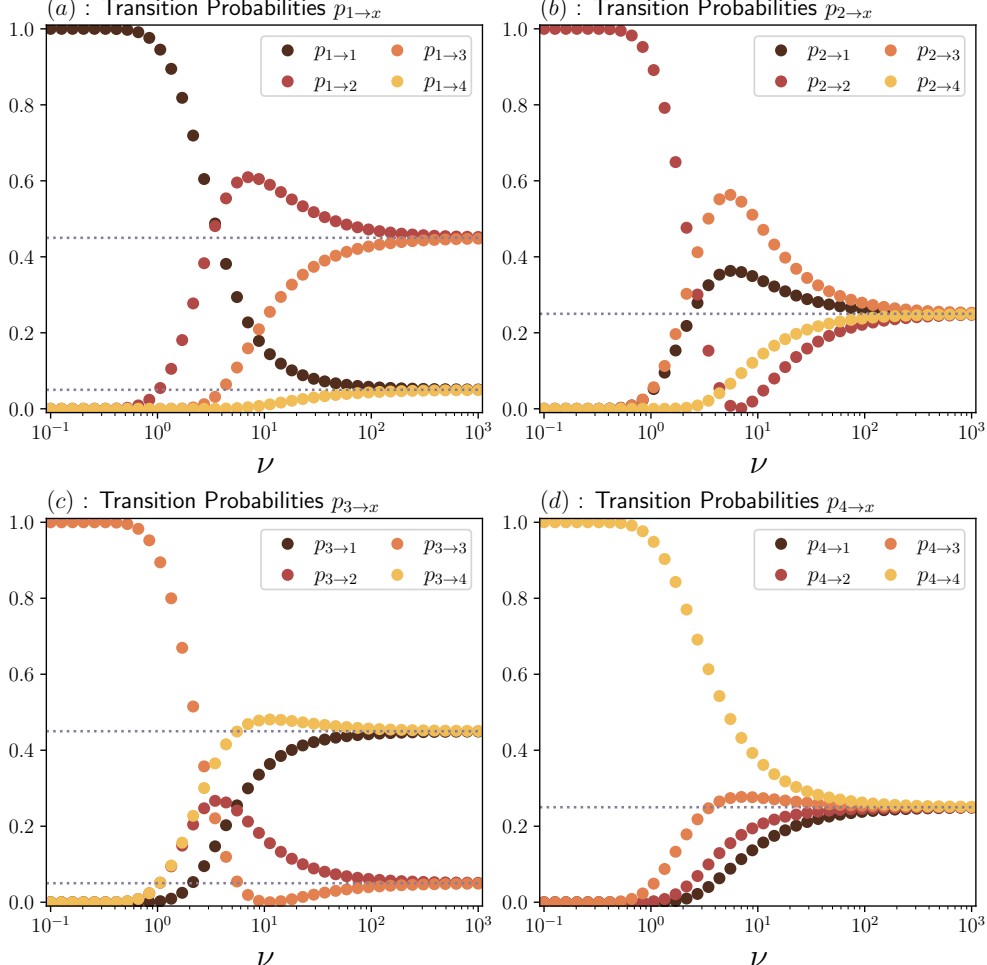

Figure 10: (a–d) Numerically evaluated transition probabilities $p_{x \to y}$ for the $4 \times 4$ HLZ model at $\tau = 10^3$ as functions of $v$ at log-equidistant values between $10^{-2}$ and $10^3$. Gray horizontal lines indicate the predicted probabilities from (89): in (a), (c) they are at $p = 1/20$ and $9/20$ and in (b), (d) at $p = 1/4$. At large $v$, the expected behaviour of $p_{x \to y}$ is verified. For $v \to 0$, $p_{x \to x}$ tends towards unity, while the rest of probabilities are approximately 0.

---

[5]Here the label '1' refers to the lowest energy eigenstate within each magnetization sector in the BCS Hamiltonian.

## 5 Conclusion & discussion

In this work, we obtained exact solutions for several hyperbolic Landau-Zener (HLZ) problems. We solved a number of nontrivial examples of the non-stationary Schrödinger equation of the form $i\frac{\partial\Psi}{\partial t} = (A + B/t)\Psi$, where $A$ and $B$ are time-independent $N \times N$ Hermitian matrices. We obtained the wavefunctions $\Psi(t)$ at arbitrary time $t$ as well as transition probabilities between eigenstates at $t = 0^+$ and $t \rightarrow +\infty$. More importantly, we demonstrated how these models arise from the generalised Knizhnik-Zamolodchikov (KZ) equations of Conformal Field Theory.

The BCS Hamiltonian with the superconducting coupling parameter inversely proportional to time shares the integrability properties of the KZ equations. By examining individual magnetization sectors in the finite-size BCS model, we readily identified plethora of integrable HLZ problems, some of which we explicitly solved, as mentioned above. Meanwhile, the KZ equations are solvable in terms of (multidimensional) contour integrals. We explicitly showed how the choice of the contours determines the initial conditions for the corresponding HLZ problem. Using the contour integral solution, two new HLZ problems were solved for the time evolution starting in the ground state, highlighting the usefulness of the KZ-HLZ connection. We suggest that most, if not all, finite-dimensional Landau-Zener problems, where the transition probabilities can be expressed in terms of elementary functions of the matrix elements of the Landau-Zener Hamiltonian, are connected to the KZ theory (including various generalisations of the KZ equations) in the manner described in this work.

There are several intriguing problems that remain unsolved. One is the general solution of the $3 \times 3$ problem (C.16), which itself is a generalisation of (52). Another problem is to identify the proper contours and to compute the integral in the solution of the KZ equations (49) corresponding to the spin-1 derived $3 \times 3$ HLZ problem. The $3 \times 3$ problem derived from a three-site spin-1/2 BCS Hamiltonian in Section 4.1 can also be entirely written in terms of contour integrals. In this work we solved for its dynamics starting from the ground state, yet the choice of contours for the evolution from excited states remain an open problem. In general, we believe that in order to systematically solve the contour integration problems described in this manuscript, cohomology methods should be employed.

As a final note, we emphasize that all models presented in this work are solely based on the $su(2)$ algebra. Generalisations of the KZ equations (and correspondingly the Gaudin magnets) to other Lie algebras should reveal new classes of integrable (H)LZ models. This is reserved as a topic for later investigation.

## Acknowledgments

The authors thank Gor Sarkissian, Jean-Sébastien Caux and Mikhail Isachenkov for helpful discussions and references.

**Code availability**   All code used to create the figures and derive results is readily available in a publicly accessible code repository [53].

**Funding information**   The work of SB and VG is partially supported by the Delta Institute for Theoretical Physics (DITP). The DITP consortium, a program of the Netherlands Organization for Scientific Research (NWO), is funded by the Dutch Ministry of Education, Culture and Science (OCW). The work of LB is partially supported by the Institute of Theoretical Physics Amsterdam (ITFA) at the University of Amsterdam (UvA). The authors also thank the Delta Institute for Theoretical Physics and the Institute of Physics at the University of Amsterdam for graciously hosting EY, which ultimately resulted in this work.

# A Contour integral solutions to KZ equations

## A.1 The $1 \times 1$ LZ-Hamiltonians

In this section we detail the contour integral solution to the Hamiltonians in equation (14). The Hamiltonians in question are

$$
\begin{aligned}
H_{1/2}^{(-1)}(\nu) &= -(\epsilon_1 + \epsilon_2), \\
H_{1/2}^{(1)}(\nu) &= (\epsilon_1 + \epsilon_2) - \frac{1}{\nu t}.
\end{aligned}
\tag{A.1}
$$

Our first observation is that the Hamiltonians in (A.1) are proportional to the single site, spin-$1/2$ ($n = 1$, $s = 1/2$) BCS Hamiltonian:

$$
H_{N=1,1/2} = 2\xi \hat{s}^z - \frac{1}{2\nu t}\hat{s}^+\hat{s}^- = 2\xi \hat{s}^z - \frac{1}{2\nu t}(1 + 2\hat{s}^z).
\tag{A.2}
$$

Specifically, (A.1) are simply equal to $2H_{n=1,1/2}$ where we identify $\xi = (\epsilon_1 + \epsilon_2)/2$. First, let us focus on the Hamiltonian $H_{1/2}^{(-1)}(\nu)$ in (A.1) which corresponds to $s^z = -1/2$ in (A.2). The Yang-Yang action (20) for (A.2) becomes

$$
S(\lambda, \xi) = -\nu t \xi.
\tag{A.3}
$$

The state $|\Phi(\lambda, \xi)\rangle$ is simply $|\downarrow\rangle$. Since $M = 0$, there is no integration to be done in equation (19) (the set of $\lambda_\alpha$ is empty), and we immediately write down the solution,

$$
\Psi(t, \epsilon) = \oint d\lambda\, e^{-\frac{i}{\nu}S(\lambda, \xi)} = e^{\left\{\frac{i}{\nu}(\nu t \xi)\right\}}|\downarrow\rangle.
\tag{A.4}
$$

This result is the same as the solution found by directly solving the non-stationary Schrödinger equation for the Hamiltonian $H_{1/2}^{(-1)}(\nu)$ in (A.1).

The Hamiltonian $H_{1/2}^{(1)}(\nu)$ in (A.1) is slightly less trivial. By directly solving the non-stationary Schrödinger equation, we find (up to normalization)

$$
\Psi(t, \xi) = e^{\left\{-\frac{i}{\nu}\left[\xi \nu t - \frac{1}{2}\log(t)\right]\right\}}|\uparrow\rangle.
\tag{A.5}
$$

For this problem, the off-shell Bethe state as defined in (21) is given by

$$
|\Phi\rangle = \prod_\alpha \hat{L}^+(\lambda_\alpha)|\downarrow\rangle = \frac{1}{(\lambda_1 - \xi)}|\uparrow\rangle.
\tag{A.6}
$$

The Yang-Yang action reads

$$
S(\lambda, \xi) = -2\nu t \xi \left(\frac{1}{2} - 1\right) + 2\nu t(\lambda_1 - \xi) + \frac{1}{2}\log(\xi - \lambda_1).
\tag{A.7}
$$

Note that we explicitly added and removed a factor of $2\nu t \xi$ in the first and second term of the action respectively. For a general one-site spin-$s$ BCS Hamiltonian one can always make a simple replacement like this. The solution is then given by

$$
\Psi(t, \xi) = \oint d\lambda\, e^{-\frac{i}{\nu}\left[\nu t \xi + 2\nu t(\lambda_1 - \xi) + \frac{1}{2}\log(\xi - \lambda_1)\right]}\frac{1}{(\lambda_1 - \xi)}|\uparrow\rangle.
\tag{A.8}
$$

Redefining $\lambda_1 - \xi \to \lambda_1$, we find

$$
\Psi(t, \xi) = e^{-i\nu t \xi}\oint d\lambda\, e^{-\frac{i}{\nu}\left[+2\nu t \lambda_1 + \frac{1}{2}\log(-\lambda_1)\right]}\frac{1}{\lambda_1}|\uparrow\rangle = e^{-i\nu t \xi}I(t).
\tag{A.9}
$$

By comparing Eqs. (A.5) and (A.9), we see that the integral $I(t)$ must equal $e^{\frac{i}{2\nu}\log(t)}$. This can be shown straightforwardly by taking the derivative of $I(t)$. Using the fact that

$$d\lambda_1 \frac{d}{dt}\left(e^{-2it\lambda_1}\right) = \frac{\lambda_1}{t} d\left(e^{-2it\lambda_1}\right), \tag{A.10}$$

we can write (removing an overall constant)

$$\frac{dI(t)}{dt} = \frac{1}{t}\oint d\left(e^{-2it\lambda_1}\right)\lambda_1^{\frac{-i}{2\nu}}. \tag{A.11}$$

Integrating by parts, we find

$$\frac{dI(t)}{dt} = \frac{1}{t}\oint d\left(e^{-2it\lambda_1}\lambda_1^{-\frac{i}{2\nu}}\right) + \frac{i}{2\nu t}\oint d\left(e^{-2it\lambda_1}\right)\lambda_1^{-\frac{i}{2\nu}-1}d\lambda_1. \tag{A.12}$$

Since the boundary term vanishes for a closed contour, we have

$$\frac{dI(t)}{dt} = \frac{i}{2\nu t}I(t) \implies I(t) \propto e^{\frac{i}{\nu}\log(t)}. \tag{A.13}$$

We end this section by noting that the calculation presented in this appendix can be generalised to an arbitrary single-site spin-$s$ BCS Hamiltonian. The calculation will be slightly more difficult, but the procedure remains the same. Let us go through the calculation below. For a single-site spin-$s$ BCS Hamiltonian of the form (A.2), we find for the Yang-Yang action:

$$S(\lambda,\xi) = 2\nu t\xi(-s+M) + 2\nu t\sum_\alpha^M\left[(\lambda_\alpha-\xi) + s\log(\xi-\lambda_\alpha) - \frac{1}{2}\sum_{\beta\neq\alpha}^M\log(\lambda_\beta-\lambda_\alpha)\right]. \tag{A.14}$$

Note that we rewrote the first two terms using $s_z = -s+M$. The off-shell Bethe state is

$$|\Phi\rangle = \prod_\alpha \hat{L}^+(\lambda_\alpha)|\downarrow\rangle = \left(\prod_\alpha^M \frac{1}{\lambda_\alpha-\xi}\right)|\uparrow\rangle. \tag{A.15}$$

The same shift as before, $\lambda_\alpha - \xi \to \lambda_\alpha$, simplifies the final result to

$$\Psi(t,\xi) = e^{-2its_z}F(t)|\uparrow\rangle, \tag{A.16}$$

where

$$F(t) = \oint d\lambda\, e^{-2it\sum_\alpha^M\left[\lambda_\alpha + s\log(-\lambda_\alpha) - \frac{1}{2}\sum_{\beta\neq\alpha}\log(\lambda_\beta-\lambda_\alpha)\right]}\prod_\alpha \lambda_\alpha^{-1}. \tag{A.17}$$

We now do the same as before: differentiate $F(t)$ and use (A.10). The steps remain largely the same, only now we have to sum over $\alpha = \{1,\ldots,M\}$ and keep track of an additional logarithmic term. We find

$$\frac{dF}{dt} = \frac{1}{t}\sum_\alpha^M\oint\prod_{\beta\neq\alpha}\left(d\lambda_\beta\lambda_\beta^{-1-\frac{is}{\nu}}e^{-2it\lambda_\beta}\right)d\left(e^{-2it\lambda_\alpha}\right)\lambda_\alpha^{-\frac{is}{\nu}}e^{\frac{i}{2\nu}\sum_{\alpha',\beta\neq\alpha'}\log(\lambda_\beta-\lambda_{\alpha'})}. \tag{A.18}$$

We integrate by parts as before,

$$\frac{dF}{dt} = -\frac{1}{B}\sum_\alpha^M\oint\prod_\beta\left(d\lambda_\beta\lambda_\beta^{-1-\frac{is}{\nu}}e^{-2it\lambda_\beta + \frac{i}{2\nu}\sum_{\beta'\neq\beta}\log(\lambda_\beta-\lambda_{\beta'})}\right)\left(-\frac{is}{\nu} + \frac{i}{\nu}\sum_{\beta\neq\alpha}\frac{\lambda_\alpha}{\lambda_\beta-\lambda_\alpha}\right). \tag{A.19}$$

The summation over $\alpha$ simplifies the second term in the square brackets,

$$\sum_\alpha \left[ -\frac{is}{\nu} + \frac{i}{\nu} \sum_{\beta \neq \alpha} \frac{\lambda_\alpha}{\lambda_\beta - \lambda_\alpha} \right] = -\frac{i}{\nu} \left[ sM - \frac{M(M-1)}{2} \right]. \tag{A.20}$$

We conclude that

$$\frac{dF(t)}{dt} = \frac{i}{t} \left[ sM - \frac{M(M-1)}{2} \right] F(t) \implies \Psi(t,\xi) = \mathcal{N} e^{-2its_z + \frac{i}{\nu}\left[ sM - \frac{M(M-1)}{2} \right]\log(t)} \lvert\uparrow\rangle, \tag{A.21}$$

where $\mathcal{N}$ is a normalization constant. This is the same result as the one we obtain through direct integration of (A.2).[6]

## A.2  The $3 \times 3$ problem involving a Kampé De Fériet function

Here we evaluate the integral

$$\psi = \oint_{\gamma_1} d\eta\, e^{2i\Delta_{31}t\eta} (\eta - r_1)^{-\frac{i}{2\nu}-1} (\eta + r_2)^{-\frac{i}{2\nu}-1} \eta^{-\frac{i}{2\nu}-1+\alpha}, \tag{A.22}$$

where the contour $\gamma_1$ is shown in Fig. 2. Notice that the singular points in the complex plane are at $0, r_1$ and $-r_2 = -(1-r_1)$ on the real axis, with $|r_1| < 1$. Using the following expansion:

$$(x+1)^{-\kappa} = \sum_{n=0}^{\infty} \frac{(-1)^n \Gamma(\kappa+n)}{n!\Gamma(\kappa)} x^{-\kappa-n}, \tag{A.23}$$

we rewrite the integral into a sum of integrals

$$\psi = \frac{1}{\Gamma(\omega)^2} \sum_{n,m=0}^{\infty} \frac{\Gamma(\omega+n)\Gamma(\omega+m)}{n!m!(-1)^m r_1^{-n} r_2^{-m}} \int_{\infty e^{i\delta}}^{(0^+)} d\eta\, e^{2i\Delta_{31}t\eta} \eta^{-3\omega-n-m+\alpha}, \tag{A.24}$$

where $\omega = 1 + i/(2\nu)$. Using the following result from [49]:

$$\int_{\infty e^{i\delta}}^{(0^+)} dz\, e^{i\Delta z} z^\alpha = \frac{2\pi i}{\Gamma(-\alpha)} \left( e^{-\frac{3\pi i}{2}} \Delta \right)^{-\alpha-1}, \qquad -\delta < \arg\Delta < \pi - \delta, \tag{A.25}$$

we obtain

$$\psi = \frac{2\pi i\, \Gamma(3\omega-\alpha)^{-1}}{(2\Delta_{31}t)^{\alpha+1-3\omega}} \sum_{n,m=0}^{\infty} \frac{1}{n!m!} \frac{\Gamma(n+\omega)}{\Gamma(\omega)} \frac{\Gamma(m+\omega)}{\Gamma(\omega)} \frac{\Gamma(3\omega-\alpha)}{\Gamma(3\omega-\alpha+n+m)} \\ \times (2i\Delta_{31}r_1 t)^n (-2i\Delta_{31}r_2 t)^m. \tag{A.26}$$

The summation is identified as a two-variable generalised hypergeometric function, namely the Kampé De Fériet function [51],

$$\psi = \frac{2\pi i\, \Gamma(3\omega-\alpha)^{-1}}{(2\Delta_{31}t)^{\alpha+1-3\omega}} F_{1:0;0}^{0:1;1} \left[ \begin{array}{c} \text{------} : \omega; \omega \\ 3\omega-\alpha : \text{---};\text{---} \end{array} \middle| \begin{array}{c} 2ir_1\Delta_{31}t \\ -2ir_2\Delta_{31}t \end{array} \right]. \tag{A.27}$$

---

[6]This can be seen straightforwardly by replacing $\hat{s}^+\hat{s}^-$ with $s(s+1) - \hat{s}_z^2 + \hat{s}_z$, resulting in a wavefunction of the form of (A.5) with a prefactor in front of the logarithm equal to the one found in (A.20).

This function simplifies for the case $r_1 = r_2$. Indeed, let us fix $n + m = p$ for some $p \geq 0$. Then, we rewrite and simplify the summation as follows,

$$
\begin{aligned}
S &= \sum_{p=0}^{\infty} \frac{\Gamma(3\omega - \alpha)(2i\Delta_{31}r_1 t)^p}{\Gamma(3\omega - \alpha + p)} \sum_{m=0}^{p} \frac{(-1)^m}{(p-m)!m!} \frac{\Gamma(p - m + \omega)}{\Gamma(\omega)} \frac{\Gamma(m + \omega)}{\Gamma(\omega)} \\
&= \sum_{p=0}^{\infty} \frac{\Gamma(3\omega - \alpha)(2i\Delta_{31}r_1 t)^p}{\Gamma(3\omega - \alpha + p)} \frac{\pi^{3/2}(-2)^p \csc(\pi\omega)}{\Gamma\left(\frac{1}{2} - \frac{p}{2}\right)\Gamma(p+1)\Gamma(\omega)\Gamma\left(-\frac{p}{2} - \omega + 1\right)} \\
&= {}_1F_2\left[{}_{\frac{3\omega}{2} - \frac{\alpha}{2}, \frac{3\omega}{2} - \frac{\alpha}{2} + \frac{1}{2}}; -(\Delta_{31}r_1 t)^2\right].
\end{aligned}
\tag{A.28}
$$

To evaluate the summations, we use identities (5.5.3), (5.5.5), and (15.2.4) in [52] as well as the following equations (see, e.g., [54]):

$$
{}_2F_1(a, b; a - b + 1; -1) = \frac{2^{-a}\sqrt{\pi}\Gamma(a - b + 1)}{\Gamma\left(\frac{a+1}{2}\right)\Gamma\left(\frac{a}{2} - b + 1\right)},
\tag{A.29}
$$

$$
{}_1F_2(a_1; b_1, b_2; z) = \sum_{k=0}^{\infty} \frac{(a_1)_k z^k}{(b_1)_k (b_2)_k k!}.
\tag{A.30}
$$

Hence for $r_1 = r_2$, we have

$$
\psi = \frac{2\pi i\,\Gamma(3\omega - \alpha)^{-1}}{(4i\Delta_{21}t)^{\alpha + 1 - 3\omega}} f_\alpha(\Delta_{21}t), \qquad f_\alpha(x) := {}_1F_2\left[{}_{\frac{3\omega}{2} - \frac{\alpha}{2}\ \frac{3\omega}{2} - \frac{\alpha}{2} + \frac{1}{2}}; -x^2\right].
\tag{A.31}
$$

Note that $r_1 = \Delta_{21}/\Delta_{31}$ and $r_2 = \Delta_{32}/\Delta_{31}$. Therefore, $r_1 = r_2$ means that $\Delta_{21} = \Delta_{32} \equiv \Delta$ which then implies that $\varepsilon_2 = \frac{1}{2}(\varepsilon_1 + \varepsilon_3)$, i.e., $\varepsilon_i$ are equally spaced. Thus, the (unnormalized) wavefunction for the time evolution starting from the ground state, which corresponds to the contour integral (44), for equally spaced $\varepsilon_i$ is

$$
|\psi_{1/2, -1/2}^1(\tau)\rangle \propto \begin{pmatrix} -\frac{1}{\sqrt{3}}\left(\frac{3}{\Gamma(3\omega - 2)}f_2(\tau) + \frac{4\tau^2}{\Gamma(3\omega)}f_0(\tau)\right) \\ +i\frac{2\sqrt{2}\tau}{\Gamma(3\omega - 1)}f_1(\tau) \\ -\frac{4\sqrt{2}\tau^2}{\Gamma(3\omega)\sqrt{3}}f_0(\tau) \end{pmatrix},
\tag{A.32}
$$

where $\tau = \Delta t$.

## A.3 The $4 \times 4$ problem involving three-variable hypergeometric function

In this section, we determine the evolution starting from the ground state for the four-site $s = 1/2, S^z = -1$ problem (80). The Yang-Yang action for this problem is

$$
S(\lambda, \varepsilon) = -\nu t(\varepsilon_1 + \varepsilon_2 + \varepsilon_3 + \varepsilon_4) + 2\nu t\lambda - \frac{1}{8}\sum_{i \neq j}^{4}\log(\varepsilon_i - \varepsilon_j) + \frac{1}{2}\sum_{i=1}^{4}\log(\varepsilon_i - \lambda).
\tag{A.33}
$$

We set $\varepsilon_1 < \varepsilon_2 < \varepsilon_3 < \varepsilon_4$ and absorb all constant terms into an overall normalization $C$. The integral representation for the solution $|\Psi(t, \varepsilon)\rangle$ reads

$$
\begin{aligned}
\frac{|\Psi(t, \varepsilon)\rangle}{C} = \Bigg[ &\oint_\gamma d\lambda\, e^{-2it\lambda}(\varepsilon_1 - \lambda)^{-\frac{i}{2\nu} - 1}(\varepsilon_2 - \lambda)^{-\frac{i}{2\nu}}(\varepsilon_3 - \lambda)^{-\frac{i}{2\nu}}(\varepsilon_4 - \lambda)^{-\frac{i}{2\nu}} |\uparrow\downarrow\downarrow\downarrow\rangle \\
&+ \oint_\gamma d\lambda\, e^{-2it\lambda}(\varepsilon_1 - \lambda)^{-\frac{i}{2\nu}}(\varepsilon_2 - \lambda)^{-\frac{i}{2\nu} - 1}(\varepsilon_3 - \lambda)^{-\frac{i}{2\nu}}(\varepsilon_4 - \lambda)^{-\frac{i}{2\nu}} |\downarrow\uparrow\downarrow\downarrow\rangle \\
&+ \oint_\gamma d\lambda\, e^{-2it\lambda}(\varepsilon_1 - \lambda)^{-\frac{i}{2\nu}}(\varepsilon_2 - \lambda)^{-\frac{i}{2\nu}}(\varepsilon_3 - \lambda)^{-\frac{i}{2\nu} - 1}(\varepsilon_4 - \lambda)^{-\frac{i}{2\nu}} |\downarrow\downarrow\uparrow\downarrow\rangle \\
&+ \oint_\gamma d\lambda\, e^{-2it\lambda}(\varepsilon_1 - \lambda)^{-\frac{i}{2\nu}}(\varepsilon_2 - \lambda)^{-\frac{i}{2\nu}}(\varepsilon_3 - \lambda)^{-\frac{i}{2\nu}}(\varepsilon_4 - \lambda)^{-\frac{i}{2\nu} - 1} |\downarrow\downarrow\downarrow\uparrow\rangle \Bigg].
\end{aligned}
\tag{A.34}
$$

In the basis

$$
\begin{aligned}
|1\rangle &= \frac{1}{2}|{\uparrow\downarrow\downarrow\downarrow}\rangle + \frac{1}{2}|{\downarrow\uparrow\downarrow\downarrow}\rangle + \frac{1}{2}|{\downarrow\downarrow\uparrow\downarrow}\rangle + \frac{1}{2}|{\downarrow\downarrow\downarrow\uparrow}\rangle\,, \\
|2\rangle &= \frac{3}{2\sqrt{5}}|{\uparrow\downarrow\downarrow\downarrow}\rangle \frac{1}{2\sqrt{5}}|{\downarrow\uparrow\downarrow\downarrow}\rangle - \frac{1}{2\sqrt{5}}|{\downarrow\downarrow\uparrow\downarrow}\rangle - \frac{3}{2\sqrt{5}}|{\downarrow\downarrow\downarrow\uparrow}\rangle\,, \\
|3\rangle &= \frac{1}{2}|{\uparrow\downarrow\downarrow\downarrow}\rangle - \frac{1}{2}|{\downarrow\uparrow\downarrow\downarrow}\rangle - \frac{1}{2}|{\downarrow\downarrow\uparrow\downarrow}\rangle + \frac{1}{2}|{\downarrow\downarrow\downarrow\uparrow}\rangle\,, \\
|4\rangle &= \frac{1}{2\sqrt{5}}|{\uparrow\downarrow\downarrow\downarrow}\rangle - \frac{3}{2\sqrt{5}}|{\downarrow\uparrow\downarrow\downarrow}\rangle + \frac{3}{2\sqrt{5}}|{\downarrow\downarrow\uparrow\downarrow}\rangle - \frac{1}{2\sqrt{5}}|{\downarrow\downarrow\downarrow\uparrow}\rangle\,,
\end{aligned}
\tag{A.35}
$$

and for $\epsilon_{n+1} = \epsilon_1 + n\Delta$ with some $\Delta \in \mathbb{R}^+$, the Hamiltonian (80) takes a tridiagonal form. Ultimately, we need to evaluate integrals of the form

$$
I_\alpha = \oint_\gamma d\lambda\, e^{-2it\lambda} \prod_{n=1}^4 (\varepsilon_n - \lambda)^{-\omega} \lambda^\alpha\,, \qquad \omega = 1 + \frac{i}{2\nu}\,, \quad \alpha = 0,1,2,3\,.
\tag{A.36}
$$

Using $\lambda = \varepsilon_4 - \eta$ and $\Delta_{ij} = \varepsilon_i - \varepsilon_j$, we reduce this integral to a weighted sum of the following integrals:

$$
J_\alpha = \oint d\eta\, e^{2it\eta} (\eta - \Delta_{41})^{-\omega} (\eta - \Delta_{42})^{-\omega} (\eta - \Delta_{43})^{-\omega} \eta^{\alpha-\omega}\,,
\tag{A.37}
$$

where $\eta^\alpha$ comes from expanding $\lambda^\alpha = (\varepsilon_4 - \eta)^\alpha$ for non-negative integer $\alpha$. Notice that we are focusing on a single power of $\eta$ from the binomial expansion of $(\varepsilon_4 - \eta)^\alpha$. Using the expansion (A.23), we find

$$
J_\alpha = \frac{1}{\Gamma(\omega)^3} \sum_{n,m,k=0}^\infty \oint d\eta\, e^{2it\eta} \frac{\Gamma(n+\omega)\Gamma(m+\omega)\Gamma(k+\omega)}{n!\,m!\,k!\,\Delta_{41}^{-n}\Delta_{42}^{-m}\Delta_{43}^{-k}} \eta^{-4\omega-n-m-k+\alpha}\,.
\tag{A.38}
$$

We then employ the Hänkel integral (A.25) and rewrite the integral as

$$
J_\alpha = \frac{2\pi i}{\Gamma(\omega)^3 (2ti)^{-4\omega+\alpha+1}} \sum_{n,m,k=0}^\infty \frac{\Gamma(n+\omega)\Gamma(m+\omega)\Gamma(k+\omega)(2ti)^{n+m+k}}{n!\,m!\,k!\,\Gamma(4\omega-\alpha+n+m+k)\Delta_{41}^{-n}\Delta_{42}^{-m}\Delta_{43}^{-k}}\,.
\tag{A.39}
$$

The above summation is the $F_B^{(3)}$ Lauricella function [51], which is a three-variable hypergeometric function. Up to an $\alpha$-independent prefactor, we have

$$
J_\alpha = \frac{1}{(2ti)^{-4\omega+\alpha+1}} F_B^{(3)}\big(\omega,\omega,\omega,-,-,-;4\omega-\alpha;2i\Delta_{41}t,2i\Delta_{42}t,2i\Delta_{43}t\big)\,.
\tag{A.40}
$$

# B    Hypergeometric expressions for a 3 × 3 LZ problem

## B.1    Evolution from excited states

The solutions to the non-stationary Schrödinger equation for the Hamiltonian (17d) starting from the excited states are as follows:

$$
|\psi_{1,0}^k(\tau)\rangle = \mathcal{N}_k\big[\phi_1^k(\tau)|1\rangle + \phi_2^k(\tau)|2\rangle + \phi_3^k(\tau)|3\rangle\big]\,, \qquad k = 2,3\,,
\tag{B.1}
$$

with

$$\phi_1^2(\tau) = \tau e^{\frac{i}{\nu}\ln(\tau)} {}_1F_2\left[{}_{\frac{1}{2}+\frac{i}{2\nu},\frac{3}{2}-\frac{i}{\nu}}^{\frac{1}{2}-\frac{i}{2\nu}}; -\tau^2\right],$$ (B.2a)

$$\phi_1^3(\tau) = \tau^2 {}_1F_2\left[{}_{\frac{3}{2}-\frac{i}{2\nu},2-\frac{3i}{2\nu}}^{1-\frac{i}{\nu}}; -\tau^2\right],$$ (B.2b)

$$\phi_2^2(\tau) = i\sqrt{3}\, e^{\frac{i}{\nu}\ln(\tau)}\left[\frac{(-\nu+2i)}{2\nu}\, {}_1F_2\left[{}_{\frac{1}{2}+\frac{i}{2\nu},\frac{3}{2}-\frac{i}{\nu}}^{\frac{1}{2}-\frac{i}{2\nu}}; -\tau^2\right]\right.$$
$$\left.+\frac{2\nu(\nu-i)\tau^2}{(\nu+i)(3\nu-2i)}\, {}_1F_2\left[{}_{\frac{3}{2}+\frac{i}{2\nu},\frac{5}{2}-\frac{i}{\nu}}^{\frac{3}{2}-\frac{i}{2\nu}}; -\tau^2\right]\right],$$ (B.2c)

$$\phi_2^3(\tau) = i\sqrt{3}\,\tau\left[\frac{(-2\nu+3i)}{2\nu}\, {}_1F_2\left[{}_{\frac{3}{2}-\frac{i}{2\nu},2-\frac{3i}{2\nu}}^{1-\frac{i}{\nu}}; -\tau^2\right]\right.$$
$$\left.+\frac{4\nu(\nu-i)\tau^2}{(3\nu-i)(4\nu-3i)}\, {}_1F_2\left[{}_{\frac{5}{2}-\frac{i}{2\nu},3-\frac{3i}{2\nu}}^{2-\frac{i}{\nu}}; -\tau^2\right]\right],$$ (B.2d)

$$\phi_3^2(\tau) = \frac{\tau e^{\frac{i}{\nu}\ln(\tau)}}{\sqrt{2}}\left[{}_1F_2\left[{}_{\frac{1}{2}+\frac{i}{2\nu},\frac{3}{2}-\frac{i}{\nu}}^{\frac{1}{2}-\frac{i}{2\nu}}; -\tau^2\right]-\frac{3(\nu-i)}{(\nu+i)}\, {}_1F_2\left[{}_{\frac{3}{2}+\frac{i}{2\nu},\frac{5}{2}-\frac{i}{\nu}}^{\frac{3}{2}-\frac{i}{2\nu}}; -\tau^2\right]\right]$$ (B.2e)
$$+\frac{6\sqrt{2}\,\nu^2(\nu-i)(3\nu-i)\tau^3 e^{\frac{i}{\nu}\ln(\tau)}}{(\nu+i)(3\nu+i)(3\nu-2i)(5\nu-2i)}\, {}_1F_2\left[{}_{\frac{5}{2}+\frac{i}{2\nu},\frac{7}{2}-\frac{i}{\nu}}^{\frac{5}{2}-\frac{i}{2\nu}}; -\tau^2\right],$$

$$\phi_3^3(\tau) = \left[\frac{3(\nu-i)(2\nu-3i)}{4\sqrt{2}\,\nu^2}+\frac{\tau^2}{\sqrt{2}}\right]{}_1F_2\left[{}_{\frac{3}{2}-\frac{i}{2\nu},2-\frac{3i}{2\nu}}^{1-\frac{i}{\nu}}; -\tau^2\right]-\frac{3\sqrt{2}(\nu-i)(5\nu-4i)\tau^2}{(3\nu-i)(4\nu-3i)}$$ (B.2f)
$$\times\, {}_1F_2\left[{}_{\frac{5}{2}-\frac{i}{2\nu},3-\frac{3i}{2\nu}}^{2-\frac{i}{\nu}}; -\tau^2\right]+\frac{8\sqrt{2}\,\nu^2(\nu-i)\tau^4}{(3\nu-i)(4\nu-3i)(5\nu-i)}\, {}_1F_2\left[{}_{\frac{7}{2}-\frac{i}{2\nu},4-\frac{3i}{2\nu}}^{3-\frac{i}{\nu}}; -\tau^2\right],$$

and

$$\mathcal{N}_2 = \left(\frac{3}{4}+\frac{3}{\nu^2}\right)^{-1/2}, \quad\text{and}\quad \mathcal{N}_3 = \left[\frac{9(\nu^2+1)(4\nu^2+9)}{32\nu^4}\right]^{-1/2}.$$ (B.2g)

This, together with the solution for the time evolution starting from the ground state, which we determined in the main text, provides the complete system of solutions for the HLZ model (17d).

## B.2 General considerations

Consider the following generalisation of differential equations in (52)

$$i\begin{pmatrix}\dot{\phi}_1(t)\\ \dot{\phi}_2(t)\\ \dot{\phi}_3(t)\end{pmatrix} = \begin{pmatrix}\frac{p}{t} & a_1 & 0\\ a_2 & \frac{q}{t} & a_3\\ 0 & a_4 & 0\end{pmatrix}\begin{pmatrix}\phi_1(t)\\ \phi_2(t)\\ \phi_3(t)\end{pmatrix}.$$ (B.3)

These differential equations are then cast into the form

$$-it^3\dddot{\phi}_1(t)+(p+q)t^2\ddot{\phi}_1(t)+\left(ipq-2p-q-i(a_4a_3+a_2a_1)t^2\right)t\dot{\phi}_1(t)$$
$$+p\left(-2iq+t^2a_3a_4+2\right)\phi_1(t) = 0,$$ (B.4a)

$$\phi_2(t) = \frac{1}{a_1}\left[i\dot{\phi}_1(t)-\frac{p}{t}\phi_1(t)\right],$$ (B.4b)

$$\phi_3(t) = \frac{1}{a_1a_3}\left[-\ddot{\phi}_1(t)-\frac{i}{t}(p+q)\dot{\phi}_1(t)+\frac{1}{t^2}(p(q+i)-t^2a_1a_2)\phi_1(t)\right].$$ (B.4c)

We only need to solve for $\phi_1(t)$ and subsequently use it to find $\phi_2(t)$ and $\phi_3(t)$. After similar simplifications, we identify the general solutions as follows

$$
\begin{aligned}
\phi_1(t) = {}& \mathcal{C}_1 t^2 {}_1F_2\left[\begin{array}{cc} 1+\frac{ipa_3a_4}{2(a_1a_2+a_3a_4)} & \\ 2+\frac{ip}{2} & \frac{3}{2}+\frac{iq}{2} \end{array} ; -\frac{1}{4}t^2\left(a_1a_2+a_3a_4\right)\right] \\
& + \mathcal{C}_2 t^{-ip} {}_1F_2\left[\begin{array}{cc} \frac{ipa_3a_4}{2(a_1a_2+a_3a_4)}-\frac{ip}{2} & \\ -\frac{1}{2}ip & \frac{i(q-p)}{2}+\frac{1}{2} \end{array} ; -\frac{1}{4}t^2\left(a_1a_2+a_3a_4\right)\right] \\
& + \mathcal{C}_3 t^{-iq+1} {}_1F_2\left[\begin{array}{cc} \frac{ipa_3a_4}{2(a_1a_2+a_3a_4)}-\frac{iq}{2}+\frac{1}{2} & \\ \frac{1}{2}-\frac{iq}{2} & \frac{i(p-q)}{2}+\frac{3}{2} \end{array} ; -\frac{1}{4}t^2\left(a_1a_2+a_3a_4\right)\right],
\end{aligned}
\tag{B.5}
$$

where $\mathcal{C}_{1,2,3}$ are arbitrary complex constants determined by the initial condition.

## C  Derivation of HLZ models from the KZ-BCS theory

In this section, we provide a step-by-step derivations of HLZ problems from an $n$-site BCS Hamiltonian for spins of magnitude $j_i$. The Hamiltonian reads

$$
H = 2\sum_{i=1}^{n}\varepsilon_i \hat{s}_i^z - \frac{1}{2vt}\left(\sum_{i=1}^{n}\hat{s}_i^+\right)\left(\sum_{i=1}^{n}\hat{s}_i^-\right),
\tag{C.1}
$$

where $s_i^z|j_i,m\rangle = m|j_i,m\rangle$ and $s_i^\pm|j_i,m\rangle = \sqrt{j_i(j_i+1)-m(m\pm1)}\,|j_i,m\pm1\rangle$.

### C.1  The $2\times2$ case

For $n=2$, we look into the $S^z = -j_1 - j_2 + 1$ sector, whose basis states are given by

$$
|b_i\rangle = \bigotimes_{j=1}^{2}|j_i, -j_i + \delta_{i,j}\rangle, \qquad i=1,2.
\tag{C.2}
$$

Investigating the matrix elements of the Hamiltonian in this basis, we determine the corresponding block

$$
H_{2\times2}^{j_1,j_2} = -2\sum_{i=1}^{2}\varepsilon_i j_i \mathbb{I} + 2\begin{pmatrix} \varepsilon_1 & 0 \\ 0 & \varepsilon_2 \end{pmatrix} - \frac{1}{vt}\begin{pmatrix} j_1 & \sqrt{j_1 j_2} \\ \sqrt{j_2 j_1} & j_2 \end{pmatrix}.
\tag{C.3}
$$

Using the unitary transformation

$$
T = \frac{1}{\sqrt{j_1 + j_2}}\begin{pmatrix} \sqrt{j_1} & \sqrt{j_2} \\ -\sqrt{j_2} & \sqrt{j_1} \end{pmatrix},
\tag{C.4}
$$

we rewrite the Hamiltonian as

$$
H_{2\times2}^{LZ} = 2\begin{pmatrix} -\frac{j_1+j_2}{2vt}-\left(\frac{j_1-j_2}{j_1+j_2}\right)(\varepsilon_2-\varepsilon_1) & (\varepsilon_2-\varepsilon_1) \\ (\varepsilon_2-\varepsilon_1) & 0 \end{pmatrix},
\tag{C.5}
$$

up to a multiple of identity $\left[\frac{2(j_2\varepsilon_1+j_1\varepsilon_2)}{j_1+j_2}-2\sum_{i=1}^{2}\varepsilon_i j_i\right]\mathbb{I}$. This model is of the form (51). Taking $j_1 = j_2$, we reproduce the parameterization (60) that we considered in the main text.

## C.2  The $3 \times 3$ cases

For $n = 3$, we are interested in the $S^z = -j_1 - j_2 - j_3 + 1$ sector. The basis states are

$$|b_i\rangle = \bigotimes_{j=1}^{3} |j_i, -j_i + \delta_{i,j}\rangle, \qquad i = 1, 2, 3. \tag{C.6}$$

In this basis,

$$H_{3\times3}^{j_1,j_2,j_3} = -2\sum_{i=1}^{3} \varepsilon_i j_i \mathbb{I} + 2 \begin{pmatrix} \varepsilon_1 & 0 & 0 \\ 0 & \varepsilon_2 & 0 \\ 0 & 0 & \varepsilon_3 \end{pmatrix} - \frac{1}{\nu t} \begin{pmatrix} j_1 & \sqrt{j_1 j_2} & \sqrt{j_1 j_3} \\ \sqrt{j_2 j_1} & j_2 & \sqrt{j_2 j_3} \\ \sqrt{j_3 j_1} & \sqrt{j_3 j_2} & j_3 \end{pmatrix}. \tag{C.7}$$

We use the following unitary transformation

$$T = \begin{pmatrix} \sqrt{\frac{j_1}{j_1+j_2+j_3}} & \sqrt{\frac{j_2}{j_1+j_2+j_3}} & \sqrt{\frac{j_3}{j_1+j_2+j_3}} \\ -\sqrt{\frac{j_2}{j_1+j_2}} & \sqrt{\frac{j_1}{j_1+j_2}} & 0 \\ -\sqrt{\frac{j_1 j_3}{(j_1+j_2)(j_1+j_2+j_3)}} & -\sqrt{\frac{j_2 j_3}{(j_1+j_2)(j_1+j_2+j_3)}} & \sqrt{\frac{j_1+j_2}{j_1+j_2+j_3}} \end{pmatrix}, \tag{C.8}$$

and choose

$$\varepsilon_2 = \frac{j_1(\varepsilon_3 - \varepsilon_1)}{j_2} + \varepsilon_3. \tag{C.9}$$

Then, the transformed model takes the form

$$H_{3\times3}^{LZ} = 2 \begin{pmatrix} -\frac{j_1+j_2+j_3}{2\nu t} & \sqrt{\frac{j_1(j_1+j_2)}{j_2(j_1+j_2+j_3)}}(\varepsilon_1 - \varepsilon_3) & 0 \\ \sqrt{\frac{j_1(j_1+j_2)}{j_2(j_1+j_2+j_3)}}(\varepsilon_1 - \varepsilon_3) & \left(\frac{j_1}{j_2}-1\right)(\varepsilon_3 - \varepsilon_1) & \sqrt{\frac{j_1 j_3}{j_2(j_1+j_2+j_3)}}(\varepsilon_1 - \varepsilon_3) \\ 0 & \sqrt{\frac{j_1 j_3}{j_2(j_1+j_2+j_3)}}(\varepsilon_1 - \varepsilon_3) & 0 \end{pmatrix}, \tag{C.10}$$

up to a multiple of identity $\left(2\varepsilon_3 - 2\sum_{i=1}^{3} \varepsilon_i j_i\right)\mathbb{I}$. The Hamiltonian (C.10) is a new integrable HLZ model of the form

$$H_{3\times3}^{LZ} = \begin{pmatrix} \frac{p}{t} & a_1 & 0 \\ a_1 & a_3 & a_2 \\ 0 & a_2 & 0 \end{pmatrix}, \tag{C.11}$$

with arbitrary real parameters $a_1, a_2, a_3$, and $p$. So far, we have not found an analytical solution to the non-stationary Schrödinger equation for this model. For $j_1 = j_2$, (C.10) takes the form of (52) with $q = 0$. The solution presented in Appendix B.2 for the $3 \times 3$ model solves the $S^z = -j_1 - j_2 - j_3 + 1$ sector of the three-site BCS model with on-site Zeeman fields $(\varepsilon_1, \frac{1}{2}(\varepsilon_1 + \varepsilon_3), \varepsilon_3)$.

We can also derive a $3 \times 3$ problem from the $n = 2$ BCS Hamiltonian. Consider the sector $S^z = -j_1 - j_2 + 2$ where $j_1, j_2 > 1/2$. The basis states are

$$|b_{n,m}\rangle = \bigotimes_{j=1}^{2} |j_i, -j_i + \delta_{n,j} + \delta_{m,j}\rangle, \quad n, m \in \{1, 2\}. \tag{C.12}$$

Choosing the ordering $(b_{1,1}, b_{1,2}, b_{2,2})$, we determine the matrix elements as

$$H_{3\times3}^{j_1,j_2} = -2\sum_{i=1}^{2} \varepsilon_i j_i \mathbb{I} + 2 \begin{pmatrix} 2\varepsilon_1 & 0 & 0 \\ 0 & \varepsilon_1 + \varepsilon_2 & 0 \\ 0 & 0 & 2\varepsilon_2 \end{pmatrix}$$
$$- \frac{1}{\nu t} \begin{pmatrix} 2j_1 + 1 & \sqrt{(2j_1 + 1)j_2} & 0 \\ \sqrt{(2j_1 + 1)j_2} & j_1 + j_2 & \sqrt{(2j_2 + 1)j_1} \\ 0 & \sqrt{(2j_2 + 1)j_1} & 2j_2 + 1 \end{pmatrix}. \tag{C.13}$$

The unitary transformation to the diabatic basis is

$$T = \begin{pmatrix} \sqrt{\frac{j_1(2j_1+1)}{(j_1+j_2)(2j_1+2j_2+1)}} & 2\sqrt{\frac{j_1 j_2}{(j_1+j_2)(2j_1+2j_2+1)}} & \sqrt{\frac{j_2(2j_2+1)}{(j_1+j_2)(2j_1+2j_2+1)}} \\ -\sqrt{\frac{(2j_1+1)j_2}{(j_1+j_2)(j_1+j_2+1)}} & \frac{j_1-j_2}{\sqrt{(j_1+j_2)(j_1+j_2+1)}} & \sqrt{\frac{j_1(2j_2+1)}{(j_1+j_2)(j_1+j_2+1)}} \\ \sqrt{\frac{j_2(2j_2+1)}{(j_1+j_2+1)(2j_1+2j_2+1)}} & -\sqrt{\frac{(2j_1+1)(2j_2+1)}{(j_1+j_2+1)(2j_1+2j_2+1)}} & \sqrt{\frac{j_1(2j_1+1)}{(j_1+j_2+1)(2j_1+2j_2+1)}} \end{pmatrix}. \tag{C.14}$$

We obtain

$$H_{3\times3,(2)}^{LZ} = 2(\varepsilon_2 - \varepsilon_1) \begin{pmatrix} \frac{(j_2-j_1)(2j_1+2j_2+1)}{(j_1+j_2)(j_1+j_2+1)} & 2\sqrt{\frac{j_1 j_2(j_1+j_2+1)}{(2j_1+2j_2+1)(j_1+j_2)^2}} & 0 \\ 2\sqrt{\frac{j_1 j_2(j_1+j_2+1)}{(2j_1+2j_2+1)(j_1+j_2)^2}} & \frac{(j_2-j_1)(j_1+j_2-1)}{(j_1+j_2)(j_1+j_2+1)} & \sqrt{\frac{(2j_1+1)(j_1+j_2)(2j_2+1)}{(2j_1+2j_2+1)(j_1+j_2+1)^2}} \\ 0 & \sqrt{\frac{(2j_1+1)(j_1+j_2)(2j_2+1)}{(2j_1+2j_2+1)(j_1+j_2+1)^2}} & 0 \end{pmatrix}$$

$$- \frac{1}{vt} \begin{pmatrix} 2j_1+2j_2+1 & 0 & 0 \\ 0 & j_1+j_2+1 & 0 \\ 0 & 0 & 0 \end{pmatrix} + 2 \left( \frac{2j_2\varepsilon_1 + 2j_1\varepsilon_2 + \varepsilon_1 + \varepsilon_2}{j_1+j_2+1} - \sum_{i=1}^{2} \varepsilon_i j_i \right) \mathbb{I}. \tag{C.15}$$

Thus, we arrive at a generalisation of the model which can be summarized as

$$H_{3\times3}^{LZ} = \begin{pmatrix} \frac{p}{t} + a_3 & a_1 & 0 \\ a_1 & \frac{q}{t} + a_4 & a_2 \\ 0 & a_2 & 0 \end{pmatrix}, \tag{C.16}$$

with arbitrary real $a_{1,2,3}$, $p$, and $q$. Thus far we have not been able to identify the general solution to this problem. Setting $j_1 = j_2$, we obtain (52). Appendix B.2 also solves the $S^z = -2j+1$ sector of the two-site BCS model with spins of the same magnitude $j$ for arbitrary, distinct on-site Zeeman fields $\varepsilon_1$ and $\varepsilon_2$.

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
