# Peer review of "Knizhnik-Zamolodchikov equations and integrable hyperbolic Landau-Zener models"

_SciPost Physics, doi:SciPost Phys. 18, 212 (2025)_

## Round 1 · Referee Report · Anonymous (Referee 1) · 2024-12-19

Strengths

1- Exact solutions to various 2x2 and 3x3 hyperbolic/Coulomb Landau-Zener (LZ) problems are presented, both for the dynamics and the final transition probabilities. 2- This works explores the interesting connection between time-dependent integrability and Landau-Zener dynamics.

Weaknesses

1- The Knizhnik-Zamolodchikov (KZ) equations are not used to obtain any new solutions, but rather to identify models which are then solved in a 'brute-force' manner. The connection between the KZ equations and the exact solutions to these LZ models hence remains somewhat unclear. 2- Section 2 reads largely as a list of mathematical expressions in terms of special (Bessel, hypergeometric) functions, with limited discussion of their derivation and interpretation.

Report

In this work the authors study Landau-Zener problems and explore their connection to the Knizhnik-Zamolodchikov equations of integrability. Following previous work by one of the authors, special classes of such Landau-Zener problems satisfy a notion of Frobenius integrability, which can be shown by embedding the LZ equation in a system of multi-time Schrodinger equations. These equations here result in the Knizhnik-Zamolodchikov equations, which admit a formal solution in terms of contour integrals.

After an introduction in Section 1, in Section 2 the authors consider two classes of 2x2 and 3x3 hyperbolic LZ problems. The exact solution for the time-dependent states and transition probabilities are given in terms of Bessel functions (2x2 problem) and hypergeometric functions (3x3 problem). In Section 3 the authors show how integrable BCS models can be used to construct integrable LZ models. For the two-level LZ problem the authors show how the formal solution to the corresponding KZ equation can be explicitly evaluated to return the known solutions of the state dynamics and transition probabilities, and discuss the difficulties when trying to extend this approach to higher-level models. Using the connection with integrable BCS models, different 3x3 and 4x4 integrable LZ models are subsequently introduced in Section 4.

The connection between LZ dynamics and KZ equations is interesting and highly nontrivial, and exact solutions to the non-stationary Schrodinger equation in a relevant setting present useful contributions to the literature. While this work is hence interesting and presents exact results, I have some reservations that make me believe this manuscript would be better suited to SciPost Physics Core or would require some major revisions for acceptance for SciPost Physics. The paper clearly meets the criteria for SciPost Physics Core, addressing a significant question and providing various exact results.

My main comment is that it is unclear how useful the KZ equations are to obtain solutions to the LZ dynamics discussed in this model. All exact solutions for the dynamics seem to be found by brute-force, and are presented without any discussion of how they are obtained or can e.g. be related to known differential equations (apart from the known case of the 2x2 model). The resulting expressions [Eqs. (15), (18), (23)] are hence difficult to interpret, and it would be useful if the authors would discuss how they arrive at these solutions and discuss the behavior of these solutions. The exact solutions also seem to be decoupled from the way these models are derived, such that it is unclear if it is the integrability that is responsible for the exact solvability of these models or if they can be mapped to a solvable differential equation in some other way (which could then extend beyond the integrable models discussed in this work). Because of this, the significance of the models introduced in Section 4 is also not so clear. If the connection between the KZ equations and the exact solutions from Section 2 would be made more explicit away from the 2x2 model, I would be happy to recommend this paper for SciPost Physics, since then the paper clearly meets the journal expectation of opening a new pathway in an existing or a new research direction.

Some minor comments - In Eq. (10) the authors mention that this model is solvable, and then focus on a specific choice of parameters. Is the model integrable/solvable for all choices of the Hamiltonian (10) or only at the specific parameters of Eq. (11)? - In Section 2.2, the authors mention that all transition probabilities need to be obtained. However, doesn't the transition probability for the 2x2 problem follow from the transition probability from the ground state using time-reversal symmetry? - In Eq. (40), the notation $\prod_{\alpha}^1 S(\lambda_{\alpha})$ is unnecessary, since the authors could simply write $S(\lambda)$. - The sentence "Due to their connection to the KZ equations, all these models are integrable, yet we will obtain exact solutions to some of them." at the start of Section 4 is somewhat confusing. - Typo: "aformentioned"

Requested changes

See report.

Recommendation

Accept in alternative Journal (see Report)

  • validity: high
  • significance: good
  • originality: high
  • clarity: good
  • formatting: good
  • grammar: perfect

Author:  Lieuwe Bakker  on 2025-06-10  [id 5557]

(in reply to Report 1 on 2024-12-19)

Dear Referee,

Thank you for your careful reading of our manuscript. We apologize for the delay in our response. We appreciate the thoughtful and important points you raised and have made every effort to address them thoroughly. Below, we respond to your comments point by point, beginning with the most significant concern you identified.

You wrote: "My main comment is that it is unclear how useful the KZ equations are to obtain solutions to the LZ dynamics discussed in this model. All exact solutions for the dynamics seem to be found by brute-force, and are presented without any discussion of how they are obtained or can e.g. be related to known differential equations (apart from the known case of the $2\times 2$ model)."

We fully agree with your assessment. We also agree that explicitly utilizing the KZ solutions would significantly strengthen our manuscript by demonstrating the practical utility of the HLZ–KZ connection. Motivated by your comment, we revisited this problem and succeeded in solving the $3\times 3$ and $4\times 4$ HLZ models for general on-site Zeeman fields by explicitly evaluating the corresponding contour integral solutions of KZ equations.

These results are new and could not be obtained by brute-force methods. In these cases, the contour integral formulation offers valuable insights into the full solutions of the associated non-stationary Schrödinger equations. Identifying and evaluating these integrals was one of the central challenges we faced, and the progress we made in this direction accounts for the delay in our resubmission. We hope these results clearly demonstrate the practical value of the HLZ–KZ correspondence established in our work.

You wrote: "The resulting expressions [Eqs. (15), (18), (23)] are hence difficult to interpret, and it would be useful if the authors would discuss how they arrive at these solutions and discuss the behavior of these solutions. "

Our response: Thank you for pointing out this important oversight. In the revised manuscript, we now provide a detailed explanation of how the solutions to the differential equations [Eqs.~(15), (18), (23)] are obtained. We agree that including these derivations is essential for clarity and reproducibility, and we hope this addition makes the results more transparent and accessible to the reader.

You wrote: "The exact solutions also seem to be decoupled from the way these models are derived, such that it is unclear if it is the integrability that is responsible for the exact solvability of these models or if they can be mapped to a solvable differential equation in some other way (which could then extend beyond the integrable models discussed in this work). Because of this, the significance of the models introduced in Section 4 is also not so clear. "

Our response: We are confident that the exact solvability of these models is a direct consequence of their KZ integrability. The KZ equations are expected to yield solutions in terms of multivariate hypergeometric functions, which reflect the underlying integrable structure. In the revised manuscript, we make this connection explicit by presenting the solutions of the $3\times 3$ and $4\times 4$ HLZ models in terms of Kampé de Fériet and Lauricella functions. These solutions, derived via the integral representations of the solutions of KZ equations, provide strong evidence that the exact solvability of the HLZ problems originates from their integrable nature.

You wrote: "If the connection between the KZ equations and the exact solutions from Section 2 would be made more explicit away from the $2\times 2$ model, I would be happy to recommend this paper for SciPost Physics, since then the paper clearly meets the journal expectation of opening a new pathway in an existing or a new research direction."

Our response: We believe that our current revision addresses the above by extending our results to higher-dimensional models in terms of multivariate hypergeometric functions. These solutions significantly broaden the applicability of the contour integral approach beyond the well-understood $2 \times 2$ case, and into previously unsolved HLZ problems such as the $3 \times 3$ and $4 \times 4$ models. We hope that this extension makes the connection between the KZ equations and exact solvability more explicit, thereby clarifying the integrable structure underlying these models and supporting the case for a new pathway in this area of research.

You wrote: "Some minor comments

  • In Eq. (10) the authors mention that this model is solvable, and then focus on a specific choice of parameters. Is the model integrable/solvable for all choices of the Hamiltonian (10) or only at the specific parameters of Eq. (11)?

  • In Section 2.2, the authors mention that all transition probabilities need to be obtained. However, doesn't the transition probability for the $2\times 2$ problem follow from the transition probability from the ground state using time-reversal symmetry?

  • In Eq. (40), the notation $\prod_{\alpha}^{1}\mathcal{S}(\lambda_\alpha)$ is unnecessary, since the authors could simply write $\mathcal{S}(λ)$.

  • The sentence 'Due to their connection to the KZ equations, all these models are integrable, yet we will obtain exact solutions to some of them.' at the start of Section 4 is somewhat confusing.

  • Typo: 'aformentioned'"

Our response: We have addressed each of your comments in the revised manuscript.

  1. Regarding Eq.~(10) (now Eq.~(51) in Section 3), the model is indeed solvable for all choices of the Hamiltonian parameters. This is now stated explicitly in the text and should also be evident from the accompanying derivations.

  2. In Section 2.2 (now Section 3.2, near the end), we have incorporated your observation within the broader discussion of the No-Go theorems studied by Sinitsyn. You are, of course, correct that symmetries—specifically time-reversal symmetry in this case—can be used to determine transition probabilities.

  3. We have removed the product notation in Eq.~(40) (now Eq.~(23)), as it was indeed unnecessary.

  4. The sentence 'Due to their connection to the KZ equations, all these models are integrable, yet we will obtain exact solutions to some of them' has been revised to: 'All these models are integrable, since they are derived from the KZ equations. However, we determine solutions in terms of known functions only for some of them.' We believe this phrasing provides better clarity.

  5. The typo 'aformentioned' has been corrected.

In order to address the concerns raised by the referee, we have significantly revised our manuscript. At a macroscopic level, we have reorganized the structure by interchanging Sections 2 and 3 to better emphasize the central message of the paper: the connection between time-dependent integrability and Landau-Zener dynamics. We summarize the major changes below for convenience.

We have changed the order of the sections to first introduce the KZ equations and associated contour integral problems in Section 2, followed by the analysis of the $2 \times 2$ and $3 \times 3$ HLZ models in Section 3.

New exact solutions for the ground state of HLZ problems derived from the 3-site and 4-site spin-1/2 BCS Hamiltonians have been added in Section 2.

Additional explanations have been included in Section 3 to clarify how the solutions to the HLZ ODEs are obtained.

New results have been presented for a previously unsolved $3 \times 3$ HLZ problem and for a new $4 \times 4$ HLZ problem in Section 4.

Appendices have been updated to reflect these revisions, including new results expressed in terms of multivariate hypergeometric functions, discussed in detail in Appendix~A.

The Introduction and Conclusion have been revised to more precisely state our claims regarding LZ solvability.

Several subsections in Sections 2 and 4 have been updated for clarity and completeness.

All figures have been redesigned to ensure compatibility with grayscale printing.

Once again, we sincerely thank the referee for their thoughtful and constructive comments.

With Kind Regards, The Authors.

---

## Round 1 · Referee Report · Anonymous (Referee 2) · 2025-1-5

Strengths

  1. The authors elucidate the connection between Landau-Zener problem and conformal field theories
  2. Explore an important concept of exact integrability in the context of quantum mechanics
  3. Introduction is very well written and as such can be useful for those who are interested in working on this or related problems

Weaknesses

  1. Section 2 is too technical. It seems that most the material presents in it can be relegated to the Appendix.

Report

Two of the senior authors of the manuscript (V. G. and E. Y.) are well known experts in mathematical physics with the history of work on Landau-Zener problem. Thus the reader should expect the in-depth and technically sound discussion. I think the quality of work is quite solid and I am sure that the manuscript will be useful for the experts working in this branch of mathematical physics. I would like to recommend this article for publication in its present form.

Recommendation

Publish (meets expectations and criteria for this Journal)

  • validity: top
  • significance: good
  • originality: top
  • clarity: high
  • formatting: good
  • grammar: excellent

Author:  Lieuwe Bakker  on 2025-06-10  [id 5556]

(in reply to Report 2 on 2025-01-05)

Dear Referee,

We thank you for your careful reading of our manuscript, and we apologize for the delay in our response to your remarks.

First and foremost, we are grateful for your positive assessment of our work.

In response to several concerns raised by Referee 1, we have made substantial revisions to the manuscript. While some technical details remain (now primarily in Section 3), we have made every effort to structure the presentation in a way that allows the reader to follow the logic of our arguments despite the complexity of the material. Where appropriate, we have moved technical derivations to the Appendix in order to improve the clarity and readability of the main text.

We hope that the revised version remains acceptable to you for publication in SciPost Physics.

Once again, we sincerely thank you for your thoughtful comments and support.

With kind regards,
The Authors

---

## Round 2 · Referee Report · Anonymous (Referee 1) · 2025-6-12

Strengths

See my previous report. Additionally, the connection between the KZ equations and Landau-Zener models is clarified and various new analytical results for the latter are presented.

Weaknesses

/

Report

I would like to thank the authors for their significant efforts in addressing my previous report. The revised version significantly clarifies the connection between the KZ equations and integrable Landau-Zener models. I agree with the authors' comments upon resubmission and am happy to recommend this paper for publication in SciPost Physics since, in my opinion, it now clearly meets the journal expectation of opening a new pathway in an existing or a new research direction (time-dependent integrability).

Requested changes

/

Recommendation

Publish (easily meets expectations and criteria for this Journal; among top 50%)

  • validity: -
  • significance: -
  • originality: -
  • clarity: -
  • formatting: -
  • grammar: -

Author:  Lieuwe Bakker  on 2025-06-20  [id 5584]

(in reply to Report 1 on 2025-06-12)

Dear Referee,

We once again thank you for your carefull assessment of the manuscript, as well as your support for the publication of our work.

With Kind Regards,
The Authors

---

## Round 2 · Author Response

Dear Editor,

We once again thank you for your efforts regarding our submission to SciPost.

We have taken slightly longer than originally intended before resubmitting our manuscript to SciPost. As you mentioned in your previous communication: "Report 1 is quite detailed, and argues persuasively that the manuscript is suitable for SciPost Physics Core, but would require major revisions for SciPost Physics." We agree with your assessment and have taken great effort to address the concerns raised by referee 1. We believe we have done so successfully. The main concern raised by Referee 1, was whether the connection between the KZ equations and the LZ models identified in our work can actually be used beyond the $2\times 2$ model described in our work.

This is a reasonable remark, as the solution of the $2\times 2$ model is known and can be obtained by other means. Thus, in an effort to showcase the applicability of the KZ equations and their solutions in terms of contour integrals, we have obtained solutions to $3\times 3$ and $4\times 4$ HLZ problems. These solutions are new, and could only be derived through the contour integral solution associated with the KZ equations. We believe that these additions to our paper should satisfy Referee 1, as they explicitly match the requirements set out by the referee: "If the connection between the KZ equations and the exact solutions from Section 2 would be made more explicit away from the 2x2 model, I would be happy to recommend this paper for SciPost Physics, since then the paper clearly meets the journal expectation of opening a new pathway in an existing or a new research direction." Of course we would patiently await Referee 1's assessment of our new additions.

We have also addressed all other comments of Referee 1.

Given that we have added substantial new material to the paper, and overall made major revisions to adress Referee 1's concerns, as well as the positive appraisal from Referee 2, we have chosen to resubmit our paper to SciPost Physics.

We look forward to your response.
With Kind Regards,
The Authors.

---

## Round 2 · List of Changes

In order to address the concerns raised by the referees, we have significantly revised our manuscript. At a macroscopic level, we have reorganized the structure by interchanging Sections 2 and 3 to better emphasize the central message of the paper: the connection between time-dependent integrability and Landau-Zener dynamics. We summarize the major changes below.

The title has been modified slightly to be more descriptive of the content of this work.

We have changed the order of the sections to first introduce the KZ equations and associated contour integral problems in Section 2, followed by the analysis of the $2 \times 2$ and $3 \times 3$ HLZ models in Section 3.

New exact solutions for the ground state of HLZ problems derived from the 3-site and 4-site spin-1/2 BCS Hamiltonians have been added in Section 2.

Additional explanations have been included in Section 3 to clarify how the solutions to the HLZ ODEs are obtained.

New results have been presented for a previously unsolved $3 \times 3$ HLZ problem and for a new $4 \times 4$ HLZ problem in Section 4.

Appendices have been updated to reflect these revisions, including new results expressed in terms of multivariate hypergeometric functions, discussed in detail in Appendix~A.

The Introduction and Conclusion have been revised to more precisely state our claims regarding LZ solvability.

Several subsections in Sections 2 and 4 have been updated for clarity and completeness.

All figures have been redesigned to ensure compatibility with grayscale printing.

---

## Editorial Decision

published